REPORT

# Vaccinia virus subverts xenophagy through phosphorylation and nuclear targeting of p62

Melanie Krause[1], Jerzy Samolej[1,2], Artur Yakimovich[1], Janos Kriston-Vizi[1], Moona Huttunen[1,2], Samuel Lara-Reyna[2], Eva-Maria Frickel[2], and Jason Mercer[1,2]

**Autophagy is an essential degradation program required for cell homeostasis. Among its functions is the engulfment and destruction of cytosolic pathogens, termed xenophagy. Not surprisingly, many pathogens use various strategies to circumvent or co-opt autophagic degradation. For poxviruses, it is known that infection activates autophagy, which however is not required for successful replication. Even though these complex viruses replicate exclusively in the cytoplasm, autophagy-mediated control of poxvirus infection has not been extensively explored. Using the prototypic poxvirus, vaccinia virus (VACV), we show that overexpression of the xenophagy receptors p62, NDP52, and Tax1Bp1 restricts poxvirus infection. While NDP52 and Tax1Bp1 were degraded, p62 initially targeted cytoplasmic virions before being shunted to the nucleus. Nuclear translocation of p62 was dependent upon p62 NLS2 and correlated with VACV kinase mediated phosphorylation of p62 T269/S272. This suggests that VACV targets p62 during the early stages of infection to avoid destruction and further implies that poxviruses exhibit multi-layered control of autophagy to facilitate cytoplasmic replication.**

## Introduction

Cell autonomous immunity represents the first line of defense used by cells to combat incoming pathogens (Randow et al., 2013). Among the various strategies employed for detection and elimination of microbial invaders is xenophagy (Levine, 2005; Nakagawa et al., 2004a; Ohsumi, 2014). This selective form of macroautophagy (Feng et al., 2014; Ravikumar et al., 2009) acts as a cytosolic defense mechanism for the detection, targeting, engulfment, and delivery of cytoplasmic pathogens to lysosomes for degradation (Dong and Levine, 2013; Orvedahl and Levine, 2008). For the most part, xenophagy uses the core autophagy machinery for the destruction of multiple pathogens and relies on a sub-class of autophagy receptors for pathogen detection, namely, the sequestome 1 (p62)–like receptors (SLRs): p62, NBR1, NDP52, Tax1Bp1, and optineurin (OPTN) (Dong and Levine, 2013; Kudchodkar and Levine, 2009).

First described for bacteria (Gutierrez et al., 2004; Nakagawa et al., 2004b; Ogawa et al., 2005), xenophagy is also known to play an antiviral role through targeted degradation of cytosolic viruses or viral components (virophagy), or via activation of other cell autonomous antiviral responses (Dong and Levine, 2013). A wide range of viruses have been reported to be targeted by xenophagy including RNA viruses such as influenza A and DNA viruses HSV-1 and HSV-2 (Dong and Levine, 2013; Lee et al., 2010; Sun et al., 2012).

Vaccinia virus (VACV) is a poxvirus, which replicates exclusively in the cytoplasm of its host cells (Condit et al., 2006; Moss, 2007). As such, it is subject to a battery of cell autonomous immune defenses (Bidgood and Mercer, 2015; Hu and Shu, 2018, 2020). To overcome this, VACV dedicates nearly half of its 200 encoded proteins to evading host cell defenses and subjugating host cell systems (Bahar et al., 2011; Bidgood and Mercer, 2015; Lu and Zhang, 2020). Cellular degradation pathways are no exception. Manipulation of the ubiquitin proteasome system, for example, has been shown to be essential for VACV genome uncoating and targeted degradation of viral restriction factors (Mercer et al., 2012; Soday et al., 2019).

Initial studies aimed at investigating sequestration of autophagic membranes for VACV assembly showed that the absence of core autophagy components, ATG5 or Beclin1, had no impact on VACV replication or infectious yield (Zhang et al., 2006). A subsequent investigation into the relationship between autophagy and VACV-membrane biogenesis reported that VACV infection results in upregulation of LC3 lipidation independent of ATG5 and ATG7 (Moloughney et al., 2011). The authors further demonstrate that VACV mediates aberrant ATG12-ATG3 conjugation and that late VACV infected cells are devoid of autophagosomes. As neither ATG3 nor LC3 lipidation were found to be required for productive infection, it was suggested that

[1]MRC Laboratory for Molecular Cell Biology, University College London, London, UK; [2]Institute of Microbiology and Infection, School of Biosciences, University of Birmingham, Birmingham, UK.

Correspondence to Jason Mercer: j.p.mercer@bham.ac.uk.

VACV might disrupt autophagy through aberrant LC3 lipidation and ATG12-ATG3 conjugation (Moloughney et al., 2011). Finally, a siRNA-based screen aimed at investigating the relevance of ATG proteins in viral replication found that VACV infection was enhanced upon depletion of SLRs important for xenophagy, including p62, NDP52, NBR1, and OPTN (Mauthe et al., 2016). Taken together, these studies suggest that VACV infection activates autophagy and that VACV either disarms or circumvents this host defense at multiple levels.

Here, we sought to gain a better understanding of how VACV circumvents xenophagy by investigating its interplay with SLRs. Overexpression of the known xenophagy receptors revealed that p62, NDP52, and Tax1Bp1 could exert some control over VACV productive infection. While VACV appears to counter the effects of NDP52 and Tax1Bp1 through targeted degradation, we found that p62 was instead relegated to the nucleus of infected cells. Molecular dissection of p62 indicated that its NLS2 was necessary and sufficient for VACV-mediated nuclear shunting. Furthermore, we show that the two VACV-encoded kinases, as opposed to cellular kinases, contribute to phosphorylation of p62 residues that serve to increase p62 NLS2 nuclear import activity. These results demonstrate that VACV actively avoids targeting by multiple SLRs during infection, uncover a potential immunomodulatory role of virus-encoded kinases, and suggest that poxviruses control xenophagy at several levels to assure successful cytoplasmic replication.

## Results and discussion

### VACV circumvents autophagic restriction through NDP52, p62, and Tax1Bp1 targeting

While the role of autophagy in VACV infection remains largely undefined, a directed siRNA screen of autophagy-related gene (ATG) proteins indicated that inhibition of autophagy was beneficial to VACV replication (Mauthe et al., 2016). The SLRs, p62, NDP52, NBR1, and OPTN, were among the strongest hits in this screen, suggesting to us that this subset of autophagy receptors may serve in the autophagic elimination of VACV. All SLRs use ubiquitin for cytosolic pathogen recognition and destruction (Deretic et al., 2013; Dupont et al., 2009; Thurston et al., 2009; Wild et al., 2011). As we previously demonstrated that VACV core proteins are ubiquitinated during cytoplasmic virus assembly, packaged into nascent virions, and ubiquitinated virus cores released into the cytoplasm during the next round of virus entry (Mercer et al., 2012), we set out to investigate the interplay among VACV, SLRs, and xenophagy.

While protein depletion studies are useful for defining cellular factors required for VACV infection (Beard et al., 2014; Kilcher et al., 2014a; Moser et al., 2010; Sivan et al., 2013), defining VACV restriction factors is often more difficult due to the virus's capacity to perturb host cell innate immune responses (Bahar et al., 2011; Bidgood and Mercer, 2015; Sivan et al., 2013). Thus, to investigate the capacity of SLRs to restrict VACV replication, we overexpressed GFP-tagged versions of the five known SLRs: NBR1, NDP52, p62, OPTN, and Tax1Bp1 (Lazarou et al., 2015). Transfected cells were infected with VACV, and the infectious yield determined at 24 h after infection (hpi).

Overexpression of NDP52, p62, and Tax1Bp1 restricted VACV production by 92%, 82%, and 75%, respectively, while overexpression of NBR1 and OPTN had no effect (Fig. 1 A). Immunoblot analysis confirmed that all SLRs were overexpressed to at least 50% of endogenous protein levels (Fig. 1 B). These results show that NDP52, p62, and Tax1Bp1 can restrict VACV infection when overexpressed, providing an indication that VACV might have an intrinsic ability to overcome SLR-mediated restriction at native expression levels.

When we monitored the endogenous levels of these SLRs during infection, we found that both NDP52 and Tax1Bp1 were reduced over time, whereas p62 protein levels remained largely unchanged during the course of infection (Fig. 1, C and D). Quantification indicated that the levels of both NDP52 and Tax1Bp1 were reduced from 8 hpi, culminating in <50% and 60%, respectively, by 24 hpi, and that p62 protein levels were not significantly reduced (Fig. 1 D). An immunofluorescence (IF) time course of cells infected with a VACV recombinant that packages an mCherry-tagged version of the core protein A5, VACV, mCH-A5 (Schmidt et al., 2011) and stained for endogenous NDP52, Tax1Bp1, or p62 confirmed the decrease in NDP52 and Tax1Bp1 protein levels compared to controls (Fig. 1 E and Fig. S1).

We reasoned that VACV either directs the cleavage or degradation of these proteins. VACV encodes two proteases, G1 and I7 (Ansarah-Sobrinho and Moss, 2004a, 2004b). While G1 substrates and cleavage specificity are unknown, I7 cleaves AGX sites in several viral structural proteins and has been shown to cleave the antiviral protein Dicer (Byrd et al., 2002; Chen et al., 2015; Novy et al., 2018). However, no AGX sites are present in either NDP52 or Tax1Bp1, and inhibition of I7 and G1 expression had no impact on reduction of NDP52 or Tax1Bp1 protein levels during infection (data not shown). These results suggested that VACV partially overcomes SLR-mediated restriction by mediating the degradation of these two SLRs. Consistent with this finding, proteomic analysis of VACV infected cells shows that NDP52 and TaxBp1 are subject to proteasome-mediated degradation (Soday et al., 2019).

As opposed these two SLRs, the IF time course showed that p62 remained stable over the course of VACV infection, but was strikingly re-localized to the nucleus (Fig. 1 E). Notably, re-localization of p62 to the nucleus also occurred in virus infected primary macrophages (Fig. S2 A). This suggested to us that VACV employs an alternative strategy to disable or evade p62.

### Nuclear targeting of p62 requires VACV early gene expression

Having shown that VACV core proteins are modified by K48-linked ubiquitin during assembly to facilitate core uncoating (Mercer et al., 2012), we reasoned that VACV might drive p62 into the nucleus to avoid ubiquitin-mediated SLR recognition and subsequent autophagic degradation. To determine whether incoming VACV cores could be recognized by p62, HeLa cells transfected with EGFP-p62 were infected with VACV mCh-A5 and subjected to live-cell imaging for 2 h starting at 1 hpi, a timepoint in which incoming viral cores are present in the cytoplasm (Rizopoulos et al., 2015). Consistent with cores being

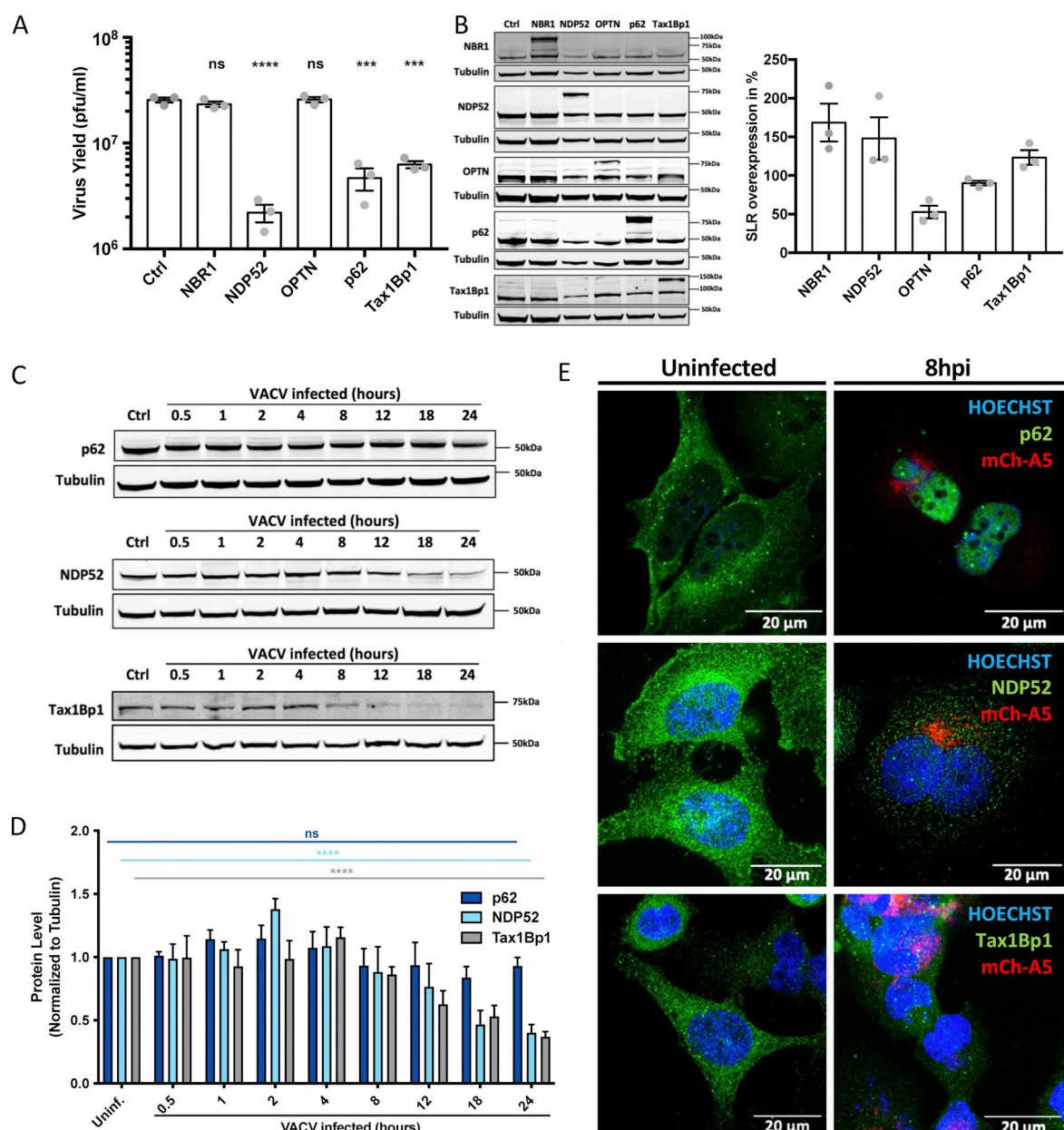

Figure 1. **VACV interferes with p62-, NDP52-, and Tax1Bp1-mediated viral restriction. (A)** 24-h yield of WT VACV from HeLa cells overexpressing the indicated SLRs. Yields were determined by plaque assay on BSC-40 cells. **(B)** Quantification of SLR overexpression at 18 h after transfection. Representative immunoblot (left) and quantification (right). **(C)** Assessment of endogenous p62, NDP52, and Tax1Bp1 protein levels during a time course of VACV infection. **(D)** Quantification of C. **(E)** Assessment of endogenous p62, NDP52, or Tax1Bp1 localization by IF in uninfected and VACV mCh-A5 infected HeLa cells at 8 hpi. SLR (green), VACV mCh-A5 (red), nuclei (blue). Representative images. A complete time course for NDP52 and Tax1Bp1 is shown in Fig. S1. All experiments were performed in triplicate ($n$ = 3) and results displayed as mean ± SEM. Unpaired two-tailed T-test with ***P ≤ 0.001 and ****P ≤ 0.0001. Scale bars = 20 μM.

ubiquitinated, instances of EGFP-p62 ring-formation around cytoplasmic VACV cores were detected by 1 hpi with co-localization lasting throughout the time course (Fig. 2 A and Fig. S2 B). We also observed incoming VACV virions within p62 ring-like structures at 2 hpi when infected cells were immunostained for p62 (Fig. S2 C). Despite finding a few instances of EGFP-p62 and endogenous p62 ring formation around cytoplasmic VACV cores, this was a rare event (<5%), which suggested to us that VACV may avoid p62 targeting during early infection.

To determine when p62 was translocated to the nucleus, we carried out an image-based time course of infection (Fig. 2 B). In uninfected cells, p62 appeared to be distributed predominantly in the cytoplasm with some present in the nucleus. The cytoplasmic signal seemed to increase at 1 hpi, before p62 appeared to aggregate in the cytoplasm by 2 hpi. By 4 hpi, p62 could be predominantly detected in the nucleus of infected cells where it then resided over the course of infection until 24 hpi, when only low levels of p62 could be detected as small aggregates in the cytoplasm (Fig. 2 B). We confirmed these results using nuclear/

Figure 2. **Incoming VACV virions are targeted by p62, which VACV evades via early gene–mediated p62 nuclear relocation. (A)** Live imaging of early-stage infection. EGFP-p62 (green)–transfected HeLa cells were infected with VACV mCh-A5 (MOI 20; red) and imaged every 10 min for 2 h, beginning 1 hpi.

Representative images of colocalization shown. **(B)** HeLa cells infected with VACV mCh-A5 (red) were fixed and immunostained for p62 (green) and DNA (blue) at the indicated timepoints. **(C)** HeLa cells, uninfected or infected WT VACV, were harvested at the indicated timepoints and subjected to nuclear/cytoplasmic fractionation followed by immunoblot analysis for endogenous p62. Tubulin and Histone 3 served as quality controls for cytoplasmic and nuclear fractionation, respectively. Quantification of nuclear/cytoplasmic endogenous p62 displayed to the right. **(D)** HeLa cells infected with VACV WR mCh-A5 in the presence of DMSO, CHX, or AraC were fixed and stained for endogenous p62 (green) and DNA (blue) at the indicated timepoints. Quantification of nuclear p62 displayed on the right. All experiments were performed in triplicate ($n = 3$) and representative images displayed. Quantifications are presented as mean ± SEM in C or ± SD in D. One-way ANOVA with ****$P ≤ 0.0001$, ***$P ≤ 0.001$, and n.s. = non-significant used in D.

cytoplasmic fractionation of infected cells combined with immunoblots for p62, as well as tubulin and Histone 3, which served as fractionation controls. Consistent with the imaging results, p62 was found mostly in the cytoplasm, but also in the nucleus of uninfected cells (Fig. 2 C). Using this as a baseline, we quantified the nuclear/cytoplasmic distribution of p62 during VACV infection (Fig. 2 C; right panel). By 1 hpi, the distribution of p62 shifted to cytoplasmic where it remained until 4 hpi. At this time, p62 distribution shifted back to the nucleus where it accumulated until 18 hpi, before equalizing between the two compartments (Fig. 2 C).

That VACV infection initiated p62 aggregation and nuclear translocation by 2 hpi suggested that a VACV early protein may be responsible for this effect. To test this, HeLa cells infected with VACV in the presence of the translation inhibitor cycloheximide (CHX) or cytosine arabinoside (AraC), which inhibits VACV DNA synthesis, were assessed for nuclear p62 (Fig. 2 D). As expected in untreated, infected cells p62 was found predominantly in the nucleus between 4 and 24 hpi. In the presence of CHX, which blocks early viral protein synthesis (Moss and Filler, 1970), p62 did not accumulate in nuclei. In the presence of AraC, nuclear accumulation of p62 was delayed (Fig. 2 D; right panel). As AraC prevents intermediate and late gene expression, without impacting early genes (Furth and Cohen, 1968), these results suggest that VACV early gene(s) initiate p62 nuclear translocation and that one or multiple late genes facilitate maintenance of this phenotype.

### Nuclear localization signal 2 (NLS2) of p62 is required for VACV-mediated nuclear shuttling

We next asked how VACV mediates nuclear translocation of p62. Of relevance, p62 contains two NLSs: NLS1 (aa 186–189) and NLS2 (aa 264–267). Mutation of the two basic residues in NLS1 or NLS2 (illustrated in Fig. 3 A) showed a 1.5- and 6.3-fold defect in nuclear import, respectively, suggesting that the NLS2 is the predominant NLS required for p62 import (Pankiv et al., 2010). To test if either NLS was important for VACV-mediated nuclear translocation, EGFP-tagged versions of p62, p62 NLS1mut, or p62 NLS2mut were transfected into HeLa cells. Immunoblot analysis assured equivalent expression levels of the three constructs (Fig. S3, A and B). Transfected cells were either left uninfected or were infected with VACV mCh-A5 and assessed for p62 nuclear translocation at 4 and 8 hpi. As expected, in uninfected cells, WT p62 appeared to be evenly distributed while both NLS1mut and NLS2mut p62 showed greater cytoplasmic accumulation (Fig. 3 B). By 4 hpi, both WT p62 and p62 NLS1mut had largely relocalized to nuclei, while the localization of p62 NLS2mut remained unchanged. A similar localization pattern was seen at 8 hpi. Quantification of the relative intensity of the nuclear EGFP

signal in VACV infected cells showed a five-fold increase in WT p62 nuclear signal by 8 hpi (Fig. 3 C). Conversely, the p62 NLS2mut showed no increase in nuclear localization upon VACV infection at either 4 or 8 hpi. The p62 NLS1mut displayed an intermediate phenotype; its nuclear signal increased 3.5-fold by 8 hpi (Fig. 3 C). We confirmed these results using nuclear/cytoplasmic fractionation of infected cells combined with immunoblots for the EGFP-tagged versions of p62, p62 NLS1mut, or p62 NLS2mut (Fig. 3 D). Consistent with previous reports, we found that mutation of NLS1 did not prevent, but impaired the efficiency of p62 nuclear shuttling (Pankiv et al., 2010). Together, these results indicated that NLS2 is required for VACV-mediated nuclear re-localization of p62 during infection.

### Cytoplasmic p62-NLS2mut reduces virus yield and localizes to VACV replication sites

Given the impact of p62 overexpression on viral yield (Fig. 1, A and B), we wondered whether the cytoplasmic retention of p62 NLS2mut would impact VACV production. For this, cells transfected with plasmids expressing EGFP, WT p62, p62 NLS1mut, or p62 NLS2mut were infected with WT VACV. At 24 hpi, the productive infectious virus yield was determined by plaque assay (Fig. 3 E). As before, over expression of WT p62 reduced virus production by 45%. While overexpression of p62 NLS1mut impacted virus yield to a similar level (50%), overexpression of p62 NLS2mut reduced virus yield an additional 27% over WT and NLS1mut p62 overexpression (Fig. 3 E).

That overexpression of the p62 NLS2mut could affect a modest reduction in virus yield suggested that cytoplasmic p62 might directly target assembling VACV virions. As VACV core proteins are ubiquitinated during assembly, we reasoned that cytoplasmic p62 would target these proteins within VACV replication sites. To accurately assign the localization of EGFP WT p62 and p62 NLS2mut in VACV infected cells, anti-EGFP antibody was used to amplify the cytoplasmic EGFP-p62 signal (Fig. 3 F). As expected, WT p62 localized to nuclei in 82% of infected cells at 8 hpi and to the cytoplasm in the other 18% (Fig. 3 G). In the case of NLS2mut, p62 was found in the cytoplasm in all cells. In 53% of these cells, p62 was predominantly associated with VACV replication sites (Fig. 3 G). Collectively these results suggest that VACV may shunt p62 to the nucleus in an attempt to prevent autophagic degradation of nascent virions.

### VACV infection upregulates p62 (T269/S272) phosphorylation independent of p38δ

Multiple studies have identified T269/S272 as two phosphorylation sites near p62 NLS2 (Nousiainen et al., 2006; Olsen et al., 2006; Pankiv et al., 2010; Yanagawa et al., 1997). Follow-up work demonstrated that a phosphomimetic p62 T269E/S272E mutant

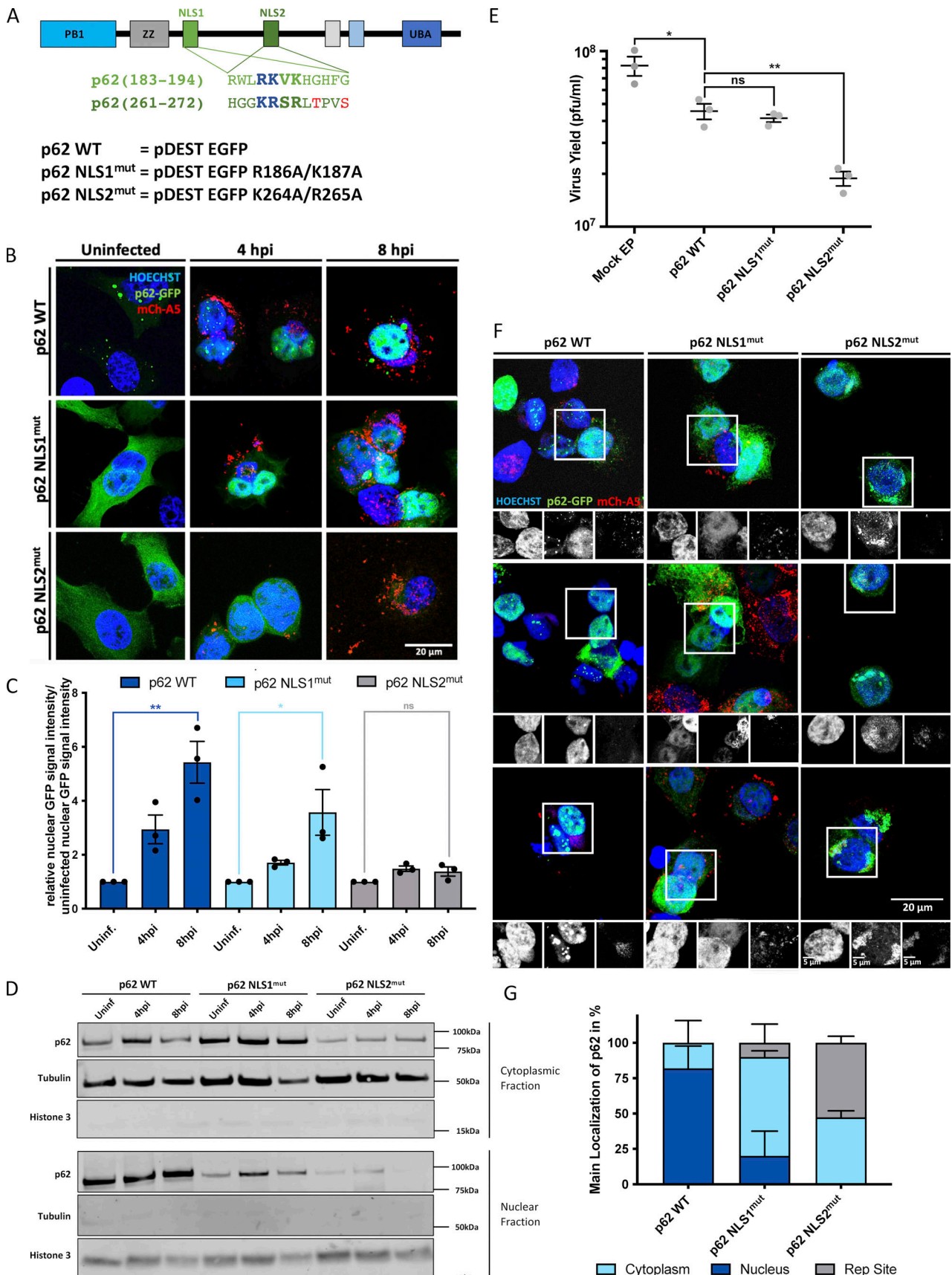

Figure 3. **VACV-mediated nuclear shunting of p62 depends on p62 NLS2. (A)** Top: Schematic representation of the two p62 NLS sites. Functional NLS motifs bolded and enlarged. Blue letters indicate amino acids changed in NLS1mut and NLS2mut. Bottom: EGFP-tagged p62 constructs WT, NLS1mut, and

NLS2[mut] with indicated mutations. UBA, ubiquitin-associated domain. **(B)** HeLa cells expressing the indicated EGFP-p62 proteins (green) were infected with VACV mCh-A5 (red) and fixed at 4 or 8 hpi. **(C)** Quantification of nuclear WT, NLS1[mut], and NLS2[mut] p62 signals from B. **(D)** Hela cells transfected with EGFP-tagged WT, NLS1[mut], or NLS2[mut] p62 were left uninfected or infected with WT VACV for 4 or 8 h. Cytosolic and nuclear cell fractions were then analyzed for p62 by immunoblot (p62-EGFP bands are shown). Immunoblots for tubulin and Histone 3 served as fractionation controls (n = 2). **(E)** 24-h yield of WT VACV from cells expressing EGFP-tagged WT, NLS1[mut], or NLS2[mut] p62. EP, electroporation. **(F)** HeLa cells expressing WT, NLS1[mut], or NLS2[mut] p62 were infected with VACV mCh-A5 (red), fixed at 8 hpi and cells immunostained with anti-GFP (green). **(G)** Quantitative classification of p62 localization from images in F. All experiments were performed in triplicate with n > 50 cells per construct per repeat and representative images displayed. Quantifications are presented as mean ± SEM. Unpaired two-tailed T-test with *P < 0.05 and **P ≤ 0.01 for C and E. A is adapted and modified from Pankiv et al. (2010).

was completely nuclear, leading the authors to conclude that T269/S272 phosphorylation serves to regulate p62 NLS2-dependent nuclear localization (Pankiv et al., 2010). Considering the robustness of VACV-mediated p62 nuclear shunting, we asked whether VACV infection increases p62 T269/S272 phosphorylation. Having seen no starvation-induced p62 phosphorylation response in HeLa cells, A549 cells were infected with WT VACV and p62 (isoform 2) T269/S272 phosphorylation assessed at various timepoints using a p62 T269/S272 phospho-specific antibody (Fig. 4 A; top). Isoform 2 lacks a portion of the PB1 domain but retains NLS1 and NLS2 including T269/S272. Increased p62 T269/S272 phosphorylation was seen by 1 hpi and remained up-regulated until 18 hpi before returning to background levels (Fig. 4 A; bottom).

Of relevance to this study, in addition to regulating p62 nuclear translocation, phosphorylation of p62 T269/S272 has been shown to be involved in mTORC1-mediated inhibition of autophagy (Linares et al., 2015). In this case, the cellular kinase p38δ phosphorylates p62 T269/S272 enabling it to recruit the E3 ubiquitin ligase TRAF6 to mTORC1 (Linares et al., 2015). Subsequent ubiquitination of mTORC1 by TRAF6 serves to inhibit autophagy (Linares et al., 2013). The increased p62 T269/S272 phosphorylation seen upon VACV infection raised the possibility that the virus may be exerting its control over p62 localization and autophagy through p38δ kinase activity.

To determine if p62 T269/S272 phosphorylation during VACV infection was via p38δ, A549 cells were either starved or infected in the absence or presence of BIRB796, an allosteric pan-p38 MAPK inhibitor (Escós et al., 2016). When p62 (isoform 2) T269/S272 phosphorylation was assessed by immunoblot, BIRB796 was found to effectively prevented starvation-induced p62 phosphorylation while no significant difference in VACV-mediated p62 phosphorylation was observed in the absence or presence of BIRB796 (Fig. 4 B). In a more targeted approach, we used siRNA-mediated depletion of p38δ. We saw no impact on p62 phosphorylation when p38δ-depleted cells were infected with VACV (Fig. S3 C). These results suggest that in our system, p38δ does not play a major role in VACV-induced p62 phosphorylation. This finding is consistent with reports that VACV-mediated dysregulation of mTORC1 protein biosynthesis activity is uncoupled from its control over autophagic responses (Meade et al., 2019).

### VACV kinases are required for p62 phosphorylation and nuclear translocation
That VACV did not appear to act through p38δ to phosphorylate p62 further suggested to us that a viral kinase may be responsible. VACV encodes two kinases, B1 and F10. B1 is a serine/

threonine kinase expressed early during infection and required for viral DNA replication (Jamin et al., 2015), and F10 is a late expressed serine/threonine/tyrosine kinase required for virus assembly (Szajner et al., 2004). As both B1 and F10 are essential for viral propagation (Lin and Broyles, 1994; Traktman et al., 1989, 1995; Wang and Shuman, 1995), recombinant viruses in which either of these genes are deleted cannot be generated. However, a B1 conditional deletion virus (ΔB1R), which only replicates in a B1R-expressing cell line, has been constructed (Olson et al., 2017).

Using the ΔB1R virus and a siRNA directed against F10 (Kilcher et al., 2014b), we asked whether either B1 or F10 contributes to p62 phosphorylation at 8 hpi, a timepoint when the F10 siRNA efficiently reduced expression of F10 protein (Fig. S3 D). Cells transfected with control or F10 siRNA were infected with WT or ΔB1R VACV and p62 (isoform 2) S269/T272 phosphorylation determined (Fig. 4 C). At 8 hpi, WT VACV infection increased p62 S269/T272 phosphorylation by 5.1-fold over uninfected cells. This was reduced to 3.2- and 3.7-fold in the absence of either F10 or B1, respectively, and further reduced to 1.9-fold when both kinases were absent.

As these results suggested that both B1 and F10 contribute to p62 phosphorylation, we next asked how loss of these kinases affects p62 nuclear translocation during VACV infection. Cells transfected with control or F10 siRNA were infected with WT or ΔB1R VACV and the cellular distribution of p62 monitored by IF at 4 and 8 hpi (Fig. 4 D; top). As expected, in the vast majority (90%) of control siRNA-transfected cells infected with WT VACV, p62 was found within the nucleus (Fig. 4 D; bottom). Depletion of F10 and deletion of B1 led to a 37% and 41% reduction in p62 nuclear translocation, respectively. In the absence of both B1 and F10, the number of cells displaying VACV-mediated p62 nuclear translocation was reduced to 21.8% (Fig. 4 D; bottom).

Collectively, these results support a role for both VACV-encoded kinases, B1 and F10, in phosphorylation of p62 and its subsequent nuclear translocation. The temporal nature of their expression (B1 early and F10 late) supports a model in which B1 initially drives p62 to the nucleus to prevent targeting of incoming ubiquitinated cores, while F10 serves to maintain this phenotype during viral morphogenesis to hamper xenophagy of nascent ubiquitinated core proteins prior to packaging. These results suggest that in addition to B1-mediated regulation of barrier-to-autointegration factor (Wiebe and Traktman, 2007), VACV-encoded kinases may play a larger role in immune modulation than previously appreciated.

In summary, using overexpression of SLRs, we have shown that xenophagy can impart control over VACV productive infection. In support of this, infection with a highly attenuated

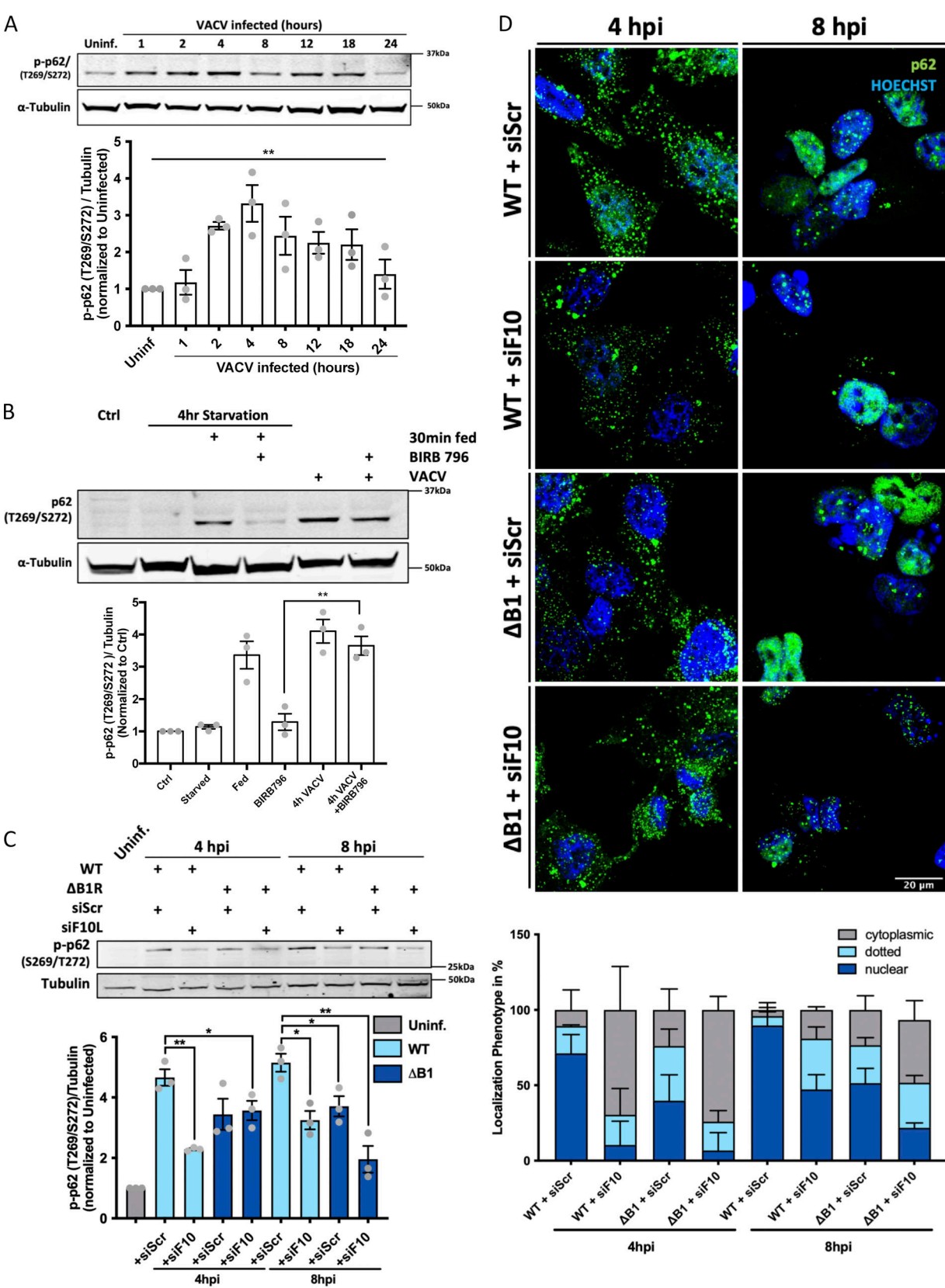

Figure 4. **VACV-encoded kinases B1 and F10 phospho-regulate p62 nuclear shunting. (A)** Top: Immunoblot analysis of p62 (isoform2) Thr269/Ser272 phosphorylation during VACV infection. Bottom: Quantification of p-p62 relative to uninfected controls. **(B)** Top: A549 cells, untreated or treated with the p38δ inhibitor BIRB796 (10 µM), were infected with VACV and p-p62 (Thr269/Ser272) assessed at 4 hpi. A549 cells starved for 4 h then fed in the absence or presence of BIRB796 served as a positive control for inhibition of p38δ-mediated p62 phosphorylation. Bottom: Quantification of p62 (Thr269/Ser272). **(C)** Top:

Immunoblot analysis of cell lysates infected with WR WT or WR ΔB1 virus in the absence or presence of VACV F10-targeting siRNA. Cells were harvested at 4 and 8 hpi and subjected to immunoblot analysis for p62 (Thr269/Ser272). Bottom: Quantification of p62 (Thr269/Ser272). **(D)** HeLa cells transfected with scrambled or F10-targeting siRNA were infected with WT or ΔB1 WR VACV, fixed at 4 or 8 hpi, and stained for p62 (green) and DNA (blue). Bottom: Quantification of p62 localization phenotype (cytoplasmic, dotted, or nuclear) at 4 and 8 hpi. All experiments were performed in triplicate and representative images are displayed. Quantifications are presented as mean ± SEM. One-way ANOVA for A and unpaired two-tailed T-test for B and C with *$P < 0.05$ and **$P \leq 0.01$.

strain of vaccinia—modified vaccinia Ankara—has been reported to result in the induction of autophagy (Tappe et al., 2018). This phenomenon is largely masked during normal infection, which we attribute to VACV's ability to effectively disarm the xenophagy receptors NDP52, p62, and Tax1Bp1. Exemplifying the ability of poxvirus to exert a multi-layered control over cell intrinsic immune responses, we show that VACV uses distinct mechanisms—degradation of NDP52 and TaxBp1 versus cytoplasmic expulsion of p62—presumably to assure that none of these SLRs can contribute to xenophagy. Interestingly, it was recently shown that nuclear p62 plays a role in trafficking of NFκB and aggresome-related proteins to nucleolar aggresomes to suppress stress-induced apoptosis (Lobb et al., 2021). Given the extensive regulation of NFκB by VACV (Smith et al., 2018), it will be of future interest to investigate possible pro-viral role(s) of nuclear p62 during poxvirus infection.

## Materials and methods

### Cells and viruses
BSC40 cells, A549 cells, and HeLa cells were maintained in Dulbecco's Modified Eagle Medium (DMEM, Life Technologies) supplemented with 10% FBS, 1 mM sodium pyruvate, 100 μM non-essential amino acids, 2 mM L-alanyl-L-glutamine dipeptide, and 1% penicillin-streptomycin at 37°C and 5% $CO_2$. Cell lines were passaged two to three times per week using PBS and Trypsin/EDTA and tested frequently for mycoplasma. VACV WR WT, WR E EGFP (Chomczynski and Mackey, 1995; Stiefel et al., 2012), WR L EGFP (Chomczynski and Mackey, 1995; Schmidt et al., 2013; Stiefel et al., 2012), and WR mCherry-A4 (Mercer and Helenius, 2008; Schmidt et al., 2013) were previously published. WR ΔB1mutB12 (Olson et al., 2019; Rico et al., 2019) was a kind gift from the Wiebe lab.

### Isolation of immune cells and macrophage differentiation
Leukocyte cones were collected from the National Health Service Blood and Transplant and processed within 4 h. Peripheral blood mononuclear cells (PBMCs) were isolated from the cones using a standard density gradient centrifugation method. After washing the samples with Dulbecco's PBS (DPBS), blood was mixed with an equal volume of DPBS (without $Ca^{2+}$ and $Mg^{2+}$, containing 2% heat-inactivated FBS) and layered onto Lymphoprep (StemCell) and centrifuged at 1,100 × $g$ for 20 min without brakes. The white buffy layer was collected and washed in DPBS by centrifuging at 300 × $g$ for 10 min. Red blood cells (RBCs) were removed by using an RBC lysis (BioLegend) for 10 min at RT. After PBMCs were obtained, CD14+ monocytes were isolated by immunomagnetic positive selection (Miltenyi Biotec).

Monocytes were seeded at a density of $0.5 \times 10^5$ and cultured in CELLview cell culture slides (Greiner).

Isolated human monocytes were cultured in complete RPMI medium containing 10% Human Serum and supplemented with 20 ng/ml human GM-CSF (PeproTech) for macrophage differentiation and incubated for 6 days, replacing the medium on day 3.

On day 6, macrophages were infected with VACV WT at MOI 50. 8 hpi cells were fixed and stained for p62 and DNA (Hoechst). Confocal fluorescence microscopy was performed using a 100× oil immersion objective (NA 1.45) on a VT-iSIM microscope (Visitech; Nikon Eclipse TI), using 405- and 488-nm laser frequencies for excitation.

### Reagents
AraC (Sigma-Aldrich) and CHX (Sigma-Aldrich) were used at 10 and 50 μM, respectively. DMSO (Sigma-Aldrich) was used to dissolve drugs and as negative control in all drug assays. p38 MAP Kinase Inhibitor BIRB 796 (MERCK Millipore) was used at 10 μM. Anti-GFP (Kilcher et al., 2014a) antibody was used at 1:500 in WB. The following antibodies were purchased from CST and used at concentrations indicated in brackets: Histone 3 (#9715S; RRID:AB_331563; western blot [WB]: 1:5,000), NDP52 (#60732S; RRID: AB_2732810; WB: 1:10,000; IF: 1:100), p-p62 (Thr269/Ser272) (#13121S; RRID: AB_2750574; WB: 1:1,000), Tax1Bp1 (#5105S; RRID: AB_11178939; WB: 1:1,000; IF: 1:100), and Tubulin (#2144S; RRID: AB_2210548; WB: 1:5,000). p62 antibody (#P0067; RRID: AB_1841064; WB: 1:5,000; IF: 1:1,000) was purchased from Sigma-Aldrich. Hoechst Trihydrochloride Trihydrate 33342 (#H3570; Invitrogen) was used for DNA staining at 1:10,000. IRDye-coupled secondary antibodies were purchased from Li-COR and used at 1:5,000. Alexa Fluor–conjugated secondary antibodies and phalloidin were purchased from Invitrogen and used at 1:400. SLR-GFP constructs were kindly gifted by Richard Youle (National Institutes of Health, Bethesda, MD, USA) (Lazarou et al., 2015). Mutant-p62 constructs were a kind gift from Terje Johansen (University of Tromsø, Tromsø, Norway) (Pankiv et al., 2010). F10L siRNA was custom-designed and manufactured by Ambion Life Technologies (F10L: 5′-GAACUACCCUGUUGCGACATT-3′; F10L_as: 5′-UGUCGCAACAGGGUAGUUCGT-3′). Mission endoribonuclease-prepared siRNA (EHU151761; Sigma-Aldrich) was used to knock down p38δ (MAPK13).

### Mature virion (MV) 24-h yield
30-mm dishes of HeLa cells were infected in DMEM-FBS at MOI 1 or MOI 0.1 with WT VACV and incubated for 1 h at 37°C. After incubation, infection media was aspirated and replaced with full DMEM, containing compounds at indicated concentrations. For siRNA-transfected cells, cells were infected at 72 h after siRNA

and incubated in full medium. After 24 h, media was aspirated, and the cells were scraped into 1 ml of PBS and spun at 300 × *g* for 5 min. Cell pellets were resuspended in 100 µl 1 mM Tris pH 9.0 and freeze-thawed three times in liquid nitrogen. The MV solution was used in a dilution series on confluent BSC40 cells for plaque assays from $10^{-4}$ to $10^{-9}$.

## Plaque assays
Plaque assays were used to determine the titer of virus stocks and 24-h yields. Serial dilutions (500 µl each) were applied to 6-well dish containing BSC-40 monolayers 500 µl of DMEM. Cells were incubated for 1 h at 37°C. Infection medium was then replaced with full DMEM containing FBS. 48 hpi at 37°C, media was aspirated and cells were fixed with 0.1% crystal violet and 2% formaldehyde. Plaque-forming units per milliliter were determined by manual counting. As BSC-40 cells are more permissive than HeLa cells, a 10-times-higher MOI is required in HeLa cells to match the infection level in BSC40 cells.

## Microscopy assays
For confocal imaging, HeLa cells were seeded on 13-mm glass coverslips (VWR) at 60,000 cells per coverslip 24 h before infection. Infection was carried out at MOI 10 in DMEM and incubated for 1 h at 37°C. The media was removed and replaced with full medium and incubated for the desired time. Cells were fixed with 4% formaldehyde-PBS for 15 min. Unless noted otherwise, cells were permeabilized with ice cold MeOH at −20°C for 20 min, blocked with 3% BSA (Sigma-Aldrich) in PBS for 1 h, and stained with 30 µl primary antibody in 3% BSA-PBS for at least 1 h at RT or overnight at 4°C. Secondary antibody staining was performed for 1 h at RT. The coverslips were mounted onto glass microscope slides. Samples were imaged using a 63× oil immersion objective (ACS APO NA 1.3) on a Leica TCS 2012 model SPE confocal microscope at RT with photomultiplier tube as detector. LAS AF was used as acquisition software. Images were processed using Fiji (RRID:SCR_002285; Fiji). For high content imaging, antibody and Hoechst staining was carried out in a 40 µl volume on a shaker. Cells were imaged at RT in 100 µl PBS. The Opera Phoenix High Content Screening System was used for image acquisition at 40× magnification with an air objective using 405-, 488-, 594-, or 647-nm lasers and at least 15 images taken per well. CellProfiler (Carpenter et al., 2006) was used to detect individual cells, based on Hoechst-stained nuclei.

## Live cell imaging
HeLa cells were transfected by electroporation with p62 WT pDEST EGFP plasmid and incubated overnight at 37°C in Cell-View cell culture slides (#543078; Greiner Bio-One). They were then infected with VACV mCh-A5 at MOI 20 and imaged live for 2 h. Imaging started at 1 hpi via confocal fluorescence microscopy with a 100× oil immersion objective (NA 1.45) on a VT-iSIM microscope (Visitech; Nikon Eclipse TI), using 488- and 561-nm laser frequencies for excitation. Multiple image locations were selected, and the stage pre-programed to automatically move and image each location every 10 min with a Z-stack of 13 images, 0.5-µm slices. All locations and timepoints were then manually analyzed for colocalization of green (p62) and red

(VACV) signal. Individual images in z processed as Z-stack of highest intensity and best co-localization have been shown here.

## Infection time courses
For WB or fractionation samples, HeLa or A549 cells were seeded in either 60-mm or 35-mm dishes for confluency at infection. Infection was carried out in DMEM without FBS at MOI 30. VACV was incubated with the cells for 1 h at 37°C, before aspirating and replacing with full medium. Cells were either untreated or treated with the indicated compounds from the start of infection and incubated at standard conditions. Samples were harvested at their respective timepoints, by removing media and washing cells with cold PBS and subsequently scraped into 50–200 µl (depending on dish size) lysis buffer containing protease inhibitor (#5872; NEB). The sample was then left on ice for at least 20 min and subsequently spun down at 20,000 × *g* for 10 min at 4°C. The supernatant was either frozen and stored at −20°C or directly supplemented with 3 × blue loading dye containing DTT. Prior to WB analysis, the samples were subject to 5-min incubation at 95°C. Protein samples were loaded into 12% Bis-Tris polyacrylamide gels (NuPAGE; Invitrogen) and transferred onto 0.2-µm nitrocellulose membrane. Membranes were blocked with 5% BSA Tris-buffered saline with Tween20 (TBS-T). Primary antibodies were applied in 5% BSA TBS-T over night at 4°C. Membranes were incubated with LiCor secondary antibodies for 1 h at RT. Imaging of membranes was carried out using a LiCor Odyssey. WB quantifications were done using ImageJ (Version2.0.0) with tubulin used as a loading control where applicable. For separation of nuclear and cytoplasmic fractions of cell lysates, the Qproteome cell compartment kit (#37502; Invitrogen) was used according to the manufacturer protocol.

## si-Knockdown experiments
siRNAs were reconstituted in sterile RNase-free water to a stock solution of 20 µM and stored at −20°C. Lipofectamine RNAiMax (Thermo Fisher Scientific) and siRNAs were diluted in pure DMEM for 5 min at RT, in volumes following the manufacturer's instructions. siRNA dilutions and media containing Lipofectamine were then mixed and incubated for a further 1 h at RT. siRNAs were used at a final concentration of 20 nM. The AllStar Hs cell death control (Quiagen) was used as a control for transfection efficiency, and AllStar negative (Quiagen) as an off-target, scrambled siRNA control. For WB analysis and confocal imaging, the siRNA-lipid complex mixture was then plated into a 30-mm dish. 300,000 A549 cells were seeded on top of the mixture in 600 µl DMEM containing FBS.

For knockdown of cellular targets, cells were incubated under standard conditions in the presence of siRNA for 72 h before infection or harvesting. For knockdown of viral targets, cells were incubated with siRNA for 24 h prior to infection with VACV. To analyze knockdown efficiency of cellular targets, cells were harvested for WB analysis as described above.

## DNA electroporation
To express mutant cellular proteins in vitro, DNA electroporation was conducted following the Amaxa Cell line nucleofector

protocol using the Cell Line Nucleofector Kit R (Lonza). 2 μg plasmid DNA was used per reaction and electroporation was carried out using the I-013 program for high expression efficiency of HeLa cells. Afterward, cells were immediately resuspended in 1.5 ml 37°C warm DMEM containing FBS and plated in a 6-well plate with one electroporation reaction per well. The cells were incubated overnight at 37°C for ~16 h before infection or harvesting for subsequent analysis.

### Statistical analysis

Statistical analyses were performed in GraphPad Prism (RRID: SCR_002798; GraphPad Prism) using unpaired two-tailed T-test or ordinary one-way ANOVA. Data distribution was assumed to be normal, but this was not formally tested. The specific tests performed are indicated in the respective figure legend. Error bars reflect SEM. The statistical tests were performed with the number of data points referring to individual biological replicates.

### Computational analysis

p62 nuclear intensity measurement (Fig. 2 D): Median filter was applied on the nuclear channel using a 3-pixel radius parameter followed by Otsu segmentation where the thresholding parameter was calculated from the image stack. A custom ImageJ macro calculated the number of nuclei using watershed segmentation algorithm with noise tolerance parameter 110.

p62 translocation measurement (Fig. 3 C): Computational analysis of the image-based data from microscopy modalities has been performed using combination of open-source software and custom-developed code. Image analysis was performed on a desktop PC equipped with Intel Core i7-8700K CPU at 3.7 GHz and 32 GB of RAM as well as GeForce 1080 Ti GPU. To measure single cell intensities in confocal image stacks, Z-maximum intensity projection was performed. Next, upon Gaussian smoothing cells were detected in a custom CellProfiler pipeline (Carpenter et al., 2006). Finally, single cell intensities were averaged per image and condition. ImageJ (Schindelin et al., 2012) (ImageJ, RRID:SCR_003070) was used to batch process the dataset.

### Online supplemental material

Fig. S1 shows the abundance of autophagy receptors NDP52 and Tax1Bp1 is reduced during VACV infection. Fig. S2 shows that VACV induces p62 nuclear re-localization in primary macrophages and incoming VACV virions can be targeted by p62. Fig. S3 shows the validation of EGFP-p62 overexpression and F10L siRNA and p38-siRNA depletion.

### Data availability

Data are available from the corresponding author upon request.

## Acknowledgments

We thank Richard J. Youle (National Institute of Neurological Disorders and Stroke, National Institutes of Health, Bethesda, MD, USA) for the autophagy SLR plasmids and cell lines, Matt Wiebe (University of Nebraska, Lincoln, NE, USA) for the ΔB1R VACV, and Terje Johansen (University of Tromsø, Tromsø, Norway) for the p62 NLS mutant plasmids.

This research was supported by MRC Laboratory for Molecular Cell Biology PhD program (M. Krause), MRC Laboratory for Molecular Cell Biology at University College London (UCL) MC_UU_00012/7 (J. Mercer), the European Research Council 649101-UbiProPox (J. Mercer), the University of Birmingham Institute of Microbiology and Infection Dynamic Investment Fund (J. Samolej/J. Mercer), and MRC core funding to the MRC-UCL University Unit Grant Ref MC_U12266B (J. Kriston-Vizi). This research was funded in part by The Wellcome Trust. A CC BY license is applied to the AAM arising from this submission, in accordance with the grant's open access conditions. E.-M. Frickel received funding by Wellcome Trust Senior Research Fellowship 217202/Z/19/Z. Open Access funding provided by University of Birmingham.

Author contributions: M. Krause contributed to conceptualization, data curation, formal analysis, funding acquisition, investigation, methodology, project administration, resources, supervision, validation, visualization, writing—original draft, and writing—review & editing. J. Samolej contributed to investigation, methodology, validation, and writing—review & editing. A. Yakimovich contributed to data curation, formal analysis, methodology, and software. J. Kriston-Vizi contributed to data curation, formal analysis, methodology, software, and writing—review & editing. M. Huttunen contributed to methodology and supervision. S. Lara-Reyna contributed to investigation. E.-M. Frickel contributed to funding acquisition, project administration, and resources. J. Mercer was involved in conceptualization, funding acquisition, methodology, project administration, supervision, validation, visualization, writing—original draft, and writing—review & editing.

Disclosures: The authors declare no competing interests exist.

Submitted: 27 April 2021

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

# Supplemental material

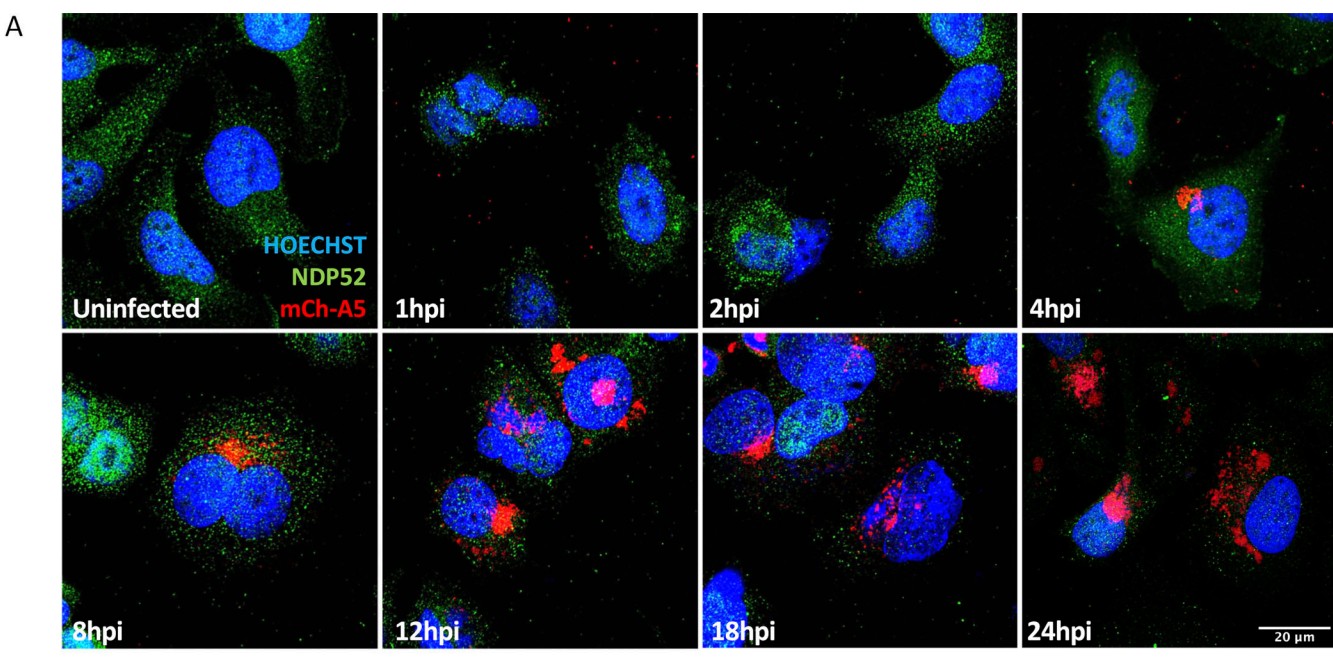

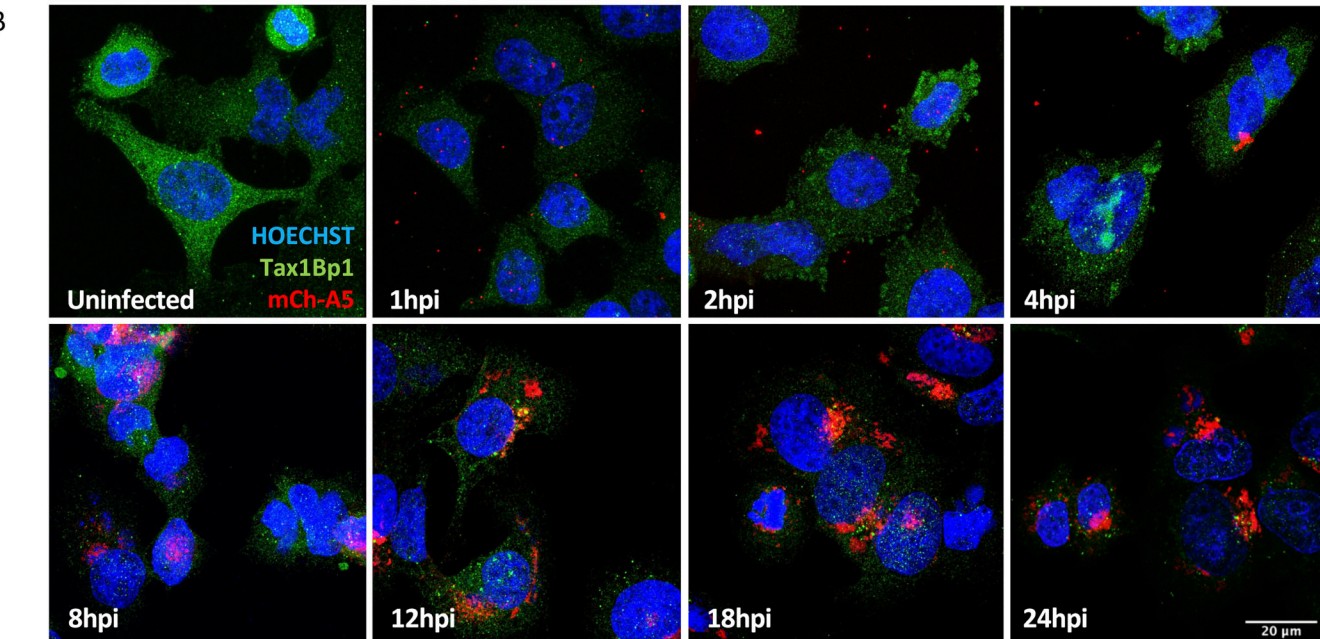

Figure S1.  **Abundance of autophagy receptors NDP52 and Tax1Bp1 is reduced during VACV infection. (A)** Time course of HeLa cells infected with WR mCh-A5 (red) and immunostained for NDP52 (green) and stained for DNA (blue). **(B)** Time course of HeLa cells infected with WR mCh-A5 (red) and immunostained for Tax1Bp1 (green) and stained for DNA (blue). 8 hpi time point also displayed in Fig. 1 E. Scale bars = 20 μM. Experiments performed in triplicate and representative images displayed.

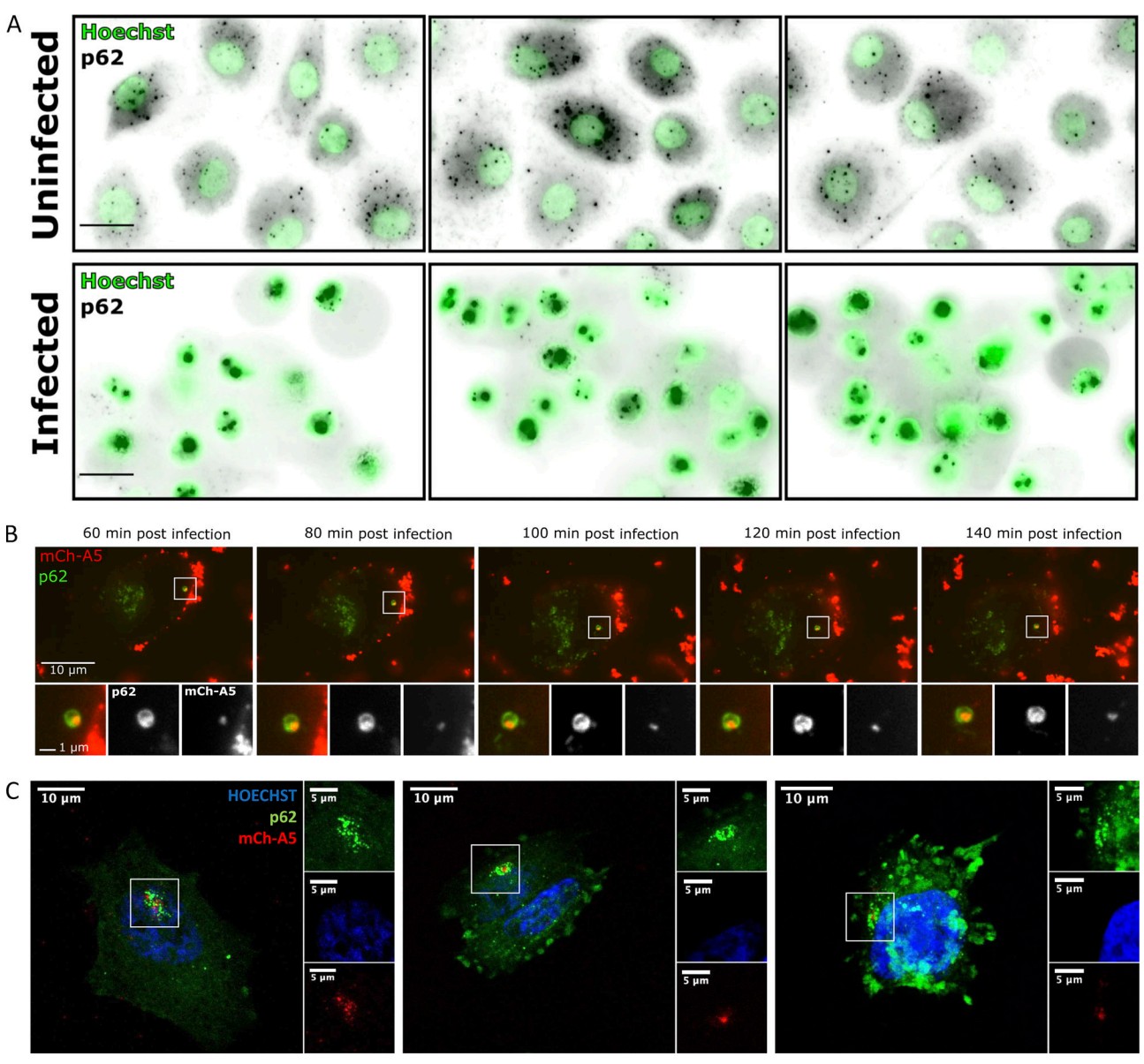

Figure S2. **VACV induces p62 nuclear re-localization in primary macrophages and incoming VACV virions can be targeted by p62. (A)** Primary macrophages were left uninfected or were infected with VACV WT (MOI 50) and fixed 8 hpi. Cells were stained for endogenous p62 (black) and DNA (green). Experiments performed in triplicate and representative images displayed. Scale bars = 10 µM. **(B)** Live-cell imaging of early-stage infection in HeLa cells. EGFP-p62 (green) –transfected HeLa cells were infected with VACV mCh-A5 (MOI 20; red) and imaged every 10 min for 2 h beginning at 1 hpi. **(C)** HeLa cells infected with VACV mCh-A5 (red) and were fixed and immunostained for p62 (green) and nuclei (blue) at 2 hpi. Scale bars = 10 µM unless stated otherwise.

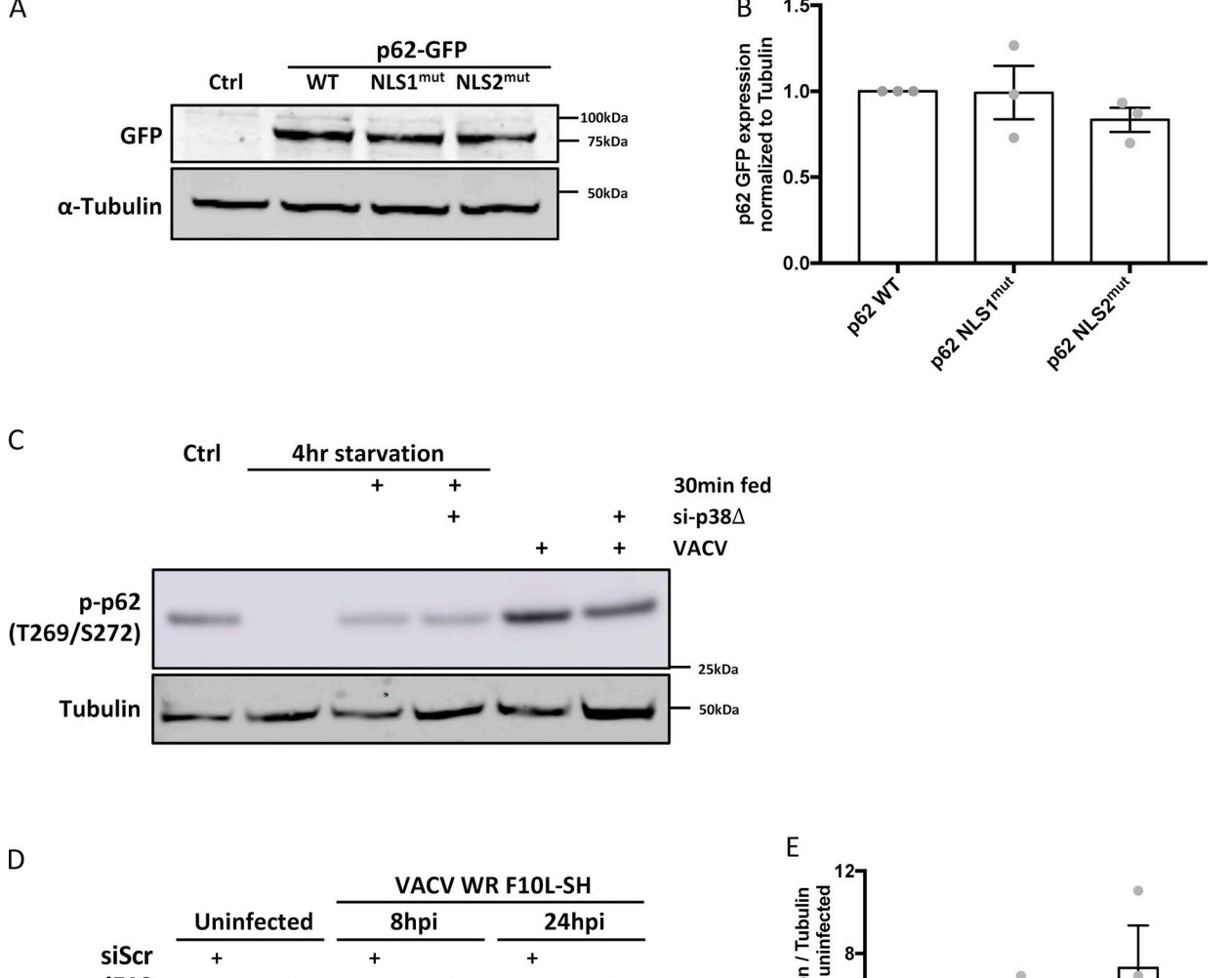

Figure S3. **Validation of EGFP-p62 overexpression and F10L siRNA and p38-siRNA depletion. (A)** Immunoblot analysis of HeLa cells expressing WT, NLS1 [mut], or NLS2 [mut] p62 at 18 h. Immunoblots directed against tubulin served for normalization. **(B)** Quantification of p62 expression displayed as mean ± SEM. Unpaired T-test. Experiment was performed in triplicate and a representative blot displayed. **(C)** A549 cells, transfected with p38δ siRNA or scrambled siRNA control, were infected with VACV and p62 (Thr269/Ser272) assessed at 4 hpi. Transfected or control siRNA cells starved for 4 h then fed served as a positive control for inhibition of p38δ-mediated p62 phosphorylation. **(D)** A549 cells transfected with either scrambled or F10-targeting siRNA were infected with VACV F10L-SH. F10 protein levels were determined at 8 and 24 hpi by immunoblot directed against the HA tag. **(E)** Quantification of F10 protein levels normalized to tubulin are displayed as mean ± SEM. Experiments were performed in triplicate and a representative blot shown.

