## [Peer Review File · The Journal of Cell Biology]

Vaccinia virus subverts xenophagy through phosphorylation and nuclear targeting of p62

Melanie Krause, Jerzy Samolej, Artur Yakimovich, Janos Kriston-Vizi, Moona Huttunen, Samuel Lara-Reyna, Eva-Maria Frickel, and Jason Mercer

Corresponding Author(s): Jason Mercer, University of Birmingham

Review Timeline:

Submission Date:	2021-04-27
Editorial Decision:	2021-06-18
Revision Received:	2023-11-13
Editorial Decision:	2024-01-17
Revision Received:	2024-01-31

Monitoring Editor: Craig Roy

Scientific Editor: Tim Spencer

Transaction Report:

DOI: <https://doi.org/10.1083/jcb.202104129>

June 18, 2021

Re: JCB manuscript #202104129

Prof. Jason Mercer
University of Birmingham
Institute of Microbiology and Infection
Edgbaston
Birmingham B15 2TT
United Kingdom

Dear Prof. Mercer,

Thank you for submitting your manuscript entitled "Vaccinia virus subverts xenophagy through phosphorylation and nuclear targeting of p62". Your manuscript has been assessed by expert reviewers, whose comments are appended below. We sincerely apologize for the delay in communicating our decision to you. Although the reviewers express potential interest in this work, they raise a substantial number of overlapping concerns regarding the solidity of the data supporting the anti-viral activity of p62 and its physiological relevance. Editorially, we agree that these are significant concerns. Unfortunately, these shortcomings reduce the reviewers' level of enthusiasm and preclude publication of the current version of the manuscript in JCB.

However, given that the work has the potential to be impactful for the field, we are willing to consider a thorough revision that address each of the reviewers' comments in full with new experimental data. In particular, we agree with reviewers 1&3 that it is essential to firmly establish the physiological relevance of p62 in this phenomenon, therefore, a successful revision would need to show that endogenous p62 displays anti-viral activity in the absence of B1 and F10 kinases. We also hope that you will be able to address each of the reviewers' other issues as well. If you are willing to do so we would be happy to consider a revised manuscript that will be reviewed by the same three experts. Please bear in mind that the JCB allows only one round of extensive revision.

Please let us know if you are able to address the major issues outlined above and wish to submit a revised manuscript to JCB. Note that a substantial amount of additional experimental data likely would be needed to satisfactorily address the concerns of the reviewers. It may be necessary to extend your manuscript to a full Research Article. As you may know, the typical timeframe for revisions is three to four months. However, we at JCB realize that the implementation of social distancing and shelter in place measures that limit spread of COVID-19 also pose challenges to scientific researchers. Lab closures especially are preventing scientists from conducting experiments to further their research. Therefore, JCB has waived the revision time limit. We recommend that you reach out to the editors once your lab has reopened to decide on an appropriate time frame for resubmission. Please note that papers are generally considered through only one revision cycle, so any revised manuscript will likely be either accepted or rejected.

If you choose to revise and resubmit your manuscript, please also attend to the following editorial points. Please direct any editorial questions to the journal office.

GENERAL GUIDELINES:

Text limits: Character count for a Report is < 20,000; a full Research Article is < 40,000, not including spaces. Count includes title page, abstract, introduction, results, discussion, acknowledgments, and figure legends. Count does not include materials and methods, references, tables, or supplemental legends.

Figures: A Report may include up to 5 main text figures; a full Research Article may have up to 10 main text figures. To avoid delays in production, figures must be prepared according to the policies outlined in our Instructions to Authors, under Data Presentation, <https://jcb.rupress.org/site/misc/ifora.xhtml>. All figures in accepted manuscripts will be screened prior to publication.

IMPORTANT: It is JCB policy that if requested, original data images must be made available. Failure to provide original images upon request will result in unavoidable delays in publication. Please ensure that you have access to all original microscopy and blot data images before submitting your revision.

Supplemental information: There are strict limits on the allowable amount of supplemental data. Reports may have up to 3 supplemental figures; a full Research Article may have up to 5 supplemental figures. Up to 10 supplemental videos or flash animations are allowed. A summary of all supplemental material should appear at the end of the Materials and methods section.

If you choose to resubmit, please include a cover letter addressing the reviewers' comments point by point. Please also highlight all changes in the text of the manuscript.

Regardless of how you choose to proceed, we hope that the comments below will prove constructive as your work progresses. We would be happy to discuss them further once you've had a chance to consider the points raised. You can contact the journal office with any questions, cellbio@rockefeller.edu.

Thank you for thinking of JCB as an appropriate place to publish your work.

Sincerely,

Craig Roy
Monitoring Editor
Journal of Cell Biology

Lucia Morgado Palacin, PhD
Scientific Editor
Journal of Cell Biology

Reviewer #1 (Comments to the Authors (Required)):

The authors report that p62 is targeted to the nucleus by two VACV kinases and that this protects the virus against host defense via xenophagy. The partial and incomplete effects on loss of function experiments for the VACV B1 and F10 kinases limit support for the interesting model. Strong evidence for this model would make a valuable contribution to our understanding of virus virulence and how xenophagy can be manipulated by viruses. For example, showing that endogenous p62 displays anti-viral activity in the absence of B1 and F10 kinases would be important.

Specific points:

- 1) Figure 1C - is this endogenous p62? If so it would help if this was stated in the results of figure legend.
- 2) A general problem is that overexpression of p62 forms clumps that make it hard to evaluate relative subcellular quantities of p62. A western blot of nuclear/cytosolic would be valuable to add to Fig. 3 as was done in Fig. 2C.
- 3) Figure 3D is not compelling. More examples and higher magnification are required to more clearly show colocalization of mutant p62 and VACV.
- 4) In Fig. 4C, p62 still gets phosphorylated following infection in the absence of B1 and knockdown of F10. This means some other kinases is also activated by VACV infection. That B1, F10 and some other kinase all phosphorylate p62 dampens enthusiasm for this proposed model of viral defense against autophagy. One key corroboration of the authors models would be to assess whether or not overexpression of B1 or F10 kinase induce endogenous p62 phosphorylation in the absence of VACV infection.
- 5) To support the imaging in Fig. 4D it would be important to show p62 location by subcellular fractionation as done in Fig. 2C.
- 6) Considering the authors work showing overexpression of NDP52, Tax1Bp1 and p62 inhibit VACV replication - what is the role of endogenous p62 in context of endogenous NDP52 and Tax1Bp1. The authors model suggests that p62 KO cells would not display a difference with WT cells because VACV inhibits endogenous p62 by phosphorylation. The key experiment then is to compare WT HeLa cells and p62 KO HeLa cells that have both been treated with F10 siRNA for replication of the B1 deleted VACV.

Reviewer #2 (Comments to the Authors (Required)):

Krause et al. report that vaccinia virus subverts xenophagy by targeting p62 to the nucleus. The authors observe that overexpression of three specific SLRs (NDP52, TAX1BP1, p62) results in (mild) reduction of viral titres and that the very same SLRs appear degraded (NDP52, TAX1BP1) or translocated to the nucleus (p62). Nuclear translocation presumably interferes with the ability of p62 to antagonize vaccinia replication. The authors report that one of the two known nuclear translocation signal in p62 (NLS2) controls nuclear translocation in vaccinia virus infected cells. They furthermore provide some evidence for two virus encoded kinases (F10 and B1) to be involved in the translocation of p62.

It is well known that autophagy antagonizes pathogen invasion and that many viruses either evade / antagonize autophagy or even take advantage of autophagy components to promote their replication. The current manuscript provides further detail to what we know about Vaccinia - autophagy interactions. However, not all data are entirely conclusive with the authors' deductions.

1. The authors conclude that VACV 'must have an intrinsic ability to overcome SLR-mediated restriction at native expression levels.' Such conclusion cannot be drawn from the overexpression experiment in Fig1.
2. The authors claim translocation of p62 into the nucleus (Fig2B). Are all images taken at the same settings? How representative are these images - quantification needed, for example percentage of p62+ nuclei.
3. If p62 translocates into the nucleus when judged by immunofluorescence (Fig2B), why is the amount of nuclear p62 not changing when judged by Western blot (Fig2C)? If p62 translocates into the nucleus via NLS2, why is there no p62 present in infected cells expressing p62 NLS2 mut at 8h p.i. (Fig3B)
4. The authors should provide information how often experiments were repeated, and whether repeats represent technical or biological replicates. Statistical analysis missing from some Figures (for example 1C) and incorrect in many others. I suggest to analyze data by ANOVA, not t-tests, for multiple comparisons (for example 1D, 2D, 3F, 4A, 4B, 4C).
5. Fig4C - shades of blue difficult to distinguish. When suggesting a role for F10 and B1, it really would be good if the authors had knockouts (and possibly double knockouts) available, as data would be much more trustworthy. Ideally complemented with wt and kinase dead mutants. Short of this, a single siRNA is insufficient to convince the reader, particularly if not complemented.
6. Fig4C - The authors comment only on the 8h time point, where they report a synergistic effect of F10 and B1. Surprisingly, at 4h pi, the late kinase F10 has a stronger phenotype than at 8h and the synergism of F10 with the early kinase B1 is weaker than at 8h pi. Why would F10 and B1, known to act early and late in the viral life cycle, respectively, appear to antagonize autophagy at opposite time points (i.e. late and early)?

Reviewer #3 (Comments to the Authors (Required)):

This manuscript provides insights into the mechanisms by which vaccinia virus (VACV) alters host cell xenophagy as a mechanism to promote replication. The authors show that infection of HeLa cells with VACV induces the specific shuttling of p62 to the nucleus very early in the viral life cycle (~2hrs post-infection). The authors go on to show that VACV infection induces the phosphorylation of p62, which the authors propose occurs due to the activity of a VACV virally-encoded kinase. Overall, the study provides new insights into the mechanisms by which VACV evades the host autophagic pathway and points to the role of VACV kinases in this process. The data are well-presented, the manuscript is well-written, and the conclusions are supported by data overall. Despite enthusiasm for the study, I do think there are instances in which additional experimentation could enhance the rigor of the work and/or provide additional clarification on key points.

1. It would appear that most of the work focuses on the overexpression of p62. It would therefore seem quite important to assess the impact of VACV infection on endogenous localization of p62.
2. All of the work was performed in HeLa cells, which have an altered autophagic pathway given their nature. The study would be improved by the inclusion of additional data from other cell types (preferably primary in nature) to support this phenomenon in other cells.
3. In Figure 2A, the authors conclude that VACV avoids p62 degradation early in infection. They base this on immunofluorescence localization studies from cells fixed at various times post-infection. As this would seem to be a key point in the manuscript, the authors should consider live-cell imaging to better resolve these early events as they may be transient in nature.
4. The effects shown in Figure 3F are quite modest. Thus, the authors may wish to alter their conclusions to better reflect these data.
5. The authors use BIRB796 to inhibit to inhibit p38 MAPK. More specific means to genetically silence or ablate expression of p38 would better support the lack of a direct link between this molecule and p62 phosphorylation and retargeting.

Reviewer #1 (Comments to the Authors (Required)):

The authors report that p62 is targeted to the nucleus by two VACV kinases and that this protects the virus against host defense via xenophagy. The partial and incomplete effects on loss of function experiments for the VACV B1 and F10 kinases limit support for the interesting model. Strong evidence for this model would make a valuable contribution to our understanding of virus virulence and how xenophagy can be manipulated by viruses. For example, showing that endogenous p62 displays anti-viral activity in the absence of B1 and F10 kinases would be important.

Specific points:

1) Figure 1C - is this endogenous p62? If so, it would help if this was stated in the results of figure legend.

Thank you for pointing this out. P62 in Fig. 1C is endogenous. We have added this information to the figure legend and main text for clarity.

2) A general problem is that overexpression of p62 forms clumps that make it hard to evaluate relative subcellular quantities of p62. A western blot of nuclear/cytosolic would be valuable to add to Fig. 3 as was done in Fig. 2C.

As suggested, we performed immunoblot analysis for EGFP-tagged versions of p62, p62 NLS1^{mut} or p62 NLS2^{mut} on the nuclear and cytoplasmic fractionation of infected cells at 4 hpi and 8 hpi. The results were consistent with panels 3B and C further suggesting the p62 NLS2 is required for VACV mediated p62 nuclear shunting. A representative immunoblot is now included as Fig 3D in the revised version of the manuscript.

3) Figure 3D is not compelling. More examples and higher magnification are required to more clearly show colocalization of mutant p62 and VACV.

To improve the visualization of Fig 3D (now Fig 3F) we have now included more examples and high-magnification split channel insets that more clearly display the overlap of p62 with VACV DNA replication sites.

4) In Fig. 4C, p62 still gets phosphorylated following infection in the absence of B1 and knockdown of F10. This means some other kinases is also activated by VACV infection. That B1, F10 and some other kinase all phosphorylate p62 dampens enthusiasm for this proposed model of viral defense against autophagy. One key corroboration of the authors models would be to assess whether or not overexpression of B1 or F10 kinase induce endogenous p62 phosphorylation in the absence of VACV infection.

Based on phosphorylation levels seen in uninfected cells in Fig 4C we are convinced that the remaining p62 phosphorylation observed in the ΔB /siF10 samples is due to incomplete knockdown of F10 kinase (See Fig S5). It has been previously shown that it is notoriously difficult to achieve complete siRNA-mediated depletion of VACV late genes due to high expression levels during infection (Rochester & Traktman, 1998).

Additionally, overexpression of active VACV encoded enzymes (F10, B1, H1) in cells in the absence of infection is VERY challenging. We have attempted to express B1 and F10 (both viral and human codon optimized versions) in many different uninfected cell lines on many occasions with little success. In addition, our previous work (Novy et al., 2018) suggests that both the molecular and

spatial interplay of F10, B1 and H1 during VACV infection are critical to their activity and substrate targeting.

Here we include unpublished independent evidence (for the reviewers) supporting the F10-mediated phosphorylation of p62. Briefly, studies have shown that T269 and S272 are phosphorylation sites near p62 NLS2 (Nousiainen et al., 2006; Olsen et al., 2006; Pankiv et al., 2010; Yanagawa et al., 1997). Follow up work demonstrated that a phosphomimetic p62 T269E/S272E mutant was completely nuclear, suggesting that T269/S272 phosphorylation regulates p62 NLS2-dependent nuclear localization (Pankiv et al., 2010).

Our lab has carried out an independent unbiased phosphoproteomic analysis of uninfected, WT VACV and WT VACV (siF10 infected cells). This analysis indicated that the phosphorylation status of p62 is altered during VACV infection. P62 T269 and S272 underwent the most significant changes, with T269 phosphorylation increasing 4-fold upon VACV infection, and both T269 and T272 phosphorylation dropping between 2- to 3- fold upon VACV infection in the absence of F10 kinase.

ptm.protein	UNIF vs. WT	p:UNIF vs. WT	F10- vs. WT	p:F10- vs. WT
Q13501_T269	1.90452143	3.60E-07	-0.780414	0.0211073
Q13501_T272	0.13122789	0.2094468	-1.4889831	1.50E-08
Q13501_T282 S283 S284 S287 S288	-0.9285179	0.03845923	-0.1022245	0.86065585
Q13501_S328	0.80652949	0.0024912	-0.87496	0.00784698
Q13501_S328 S332	0.95855528	0.02715204	0.5390391	0.25921898
Q13501_S328 S332 T339	1.32584436	0.02242655	0.18032309	0.79503664
Q13501_S332	0.86232019	1.29E-06	-0.6757387	0.00015504
Q13501_S361 S365 S366	0.08555935	0.8435494	-0.9111011	0.09079957
Q13501_S361 S365 S366 S370 T375	-0.2190645	0.29412267	-0.8354943	0.01260497
Q13501_S365 S366 T375	-0.1523459	0.64381922	-0.6577614	0.11527161

5) To support the imaging in Fig. 4D it would be important to show p62 location by subcellular fractionation as done in Fig. 2C.

In line with the reviewers request we attempted cell fractionation and immunoblot analysis of p62 on these experimental samples multiple times. The variable level of p62 recovered after fractionation did not allow for interpretation of the immunoblots. We suspect this is due to the complex nature of the p62 phenotype we observed in this experiment (cytoplasmic/nuclear/dotted). We believe the imaging experiment, as presented, is still informative as it allows us to observe and quantify these phenotypes, which support a role for F10 and B1 kinases in P62 nuclear shunting.

6) Considering the authors work showing overexpression of NDP52, Tax1Bp1 and p62 inhibit VACV replication - what is the role of endogenous p62 in context of endogenous NDP52 and Tax1Bp1. The authors model suggests that p62 KO cells would not display a difference with WT cells because VACV inhibits endogenous p62 by phosphorylation. The key experiment then is to compare WT HeLa cells and p62 KO HeLa cells that have both been treated with F10 siRNA for replication of the B1 deleted VACV.

This is correct. We obtained p62^{-/-} cells and did not observe any difference in 24-hour yield or plaque formation (data included for here for the reviewers)

A) HeLa WT and HeLa p62 KO cells grown in 30mm dishes were harvested and lysed, and the lysate analysed via western blotting, using the anti-p62 antibody. B) Example of a 24h yield plaque assay. WT and p62 KO cells were infected with VACV WT at an MOI of 1 in 6 well plates. 24 hpi cells were harvested and lysed by freeze-thawing. The lysate was serially diluted and confluent BSC40 cells were used for a 48h plaque assay. C) Quantification of plaque assays of 3 independent 24h yields described in B). Error bars are S.D.

However, the key experiment suggested by the reviewer is technically impossible. Both B1 and F10 are essential VACV kinases that play multiple roles in the virus lifecycle. Deletion of B1 prevents VACV replication while deletion of F10 blocks VACV morphogenesis and virion formation. This makes it impossible to use conventional virological or image based VACV fitness readouts (yield, DNA replication, virion assembly) as the virus lifecycle is attenuated irrespective of the p62 status of the cells.

Reviewer #2 (Comments to the Authors (Required)):

Krause et al. report that vaccinia virus subverts xenophagy by targeting p62 to the nucleus. The authors observe that overexpression of three specific SLRs (NDP52, TAX1BP1, p62) results in (mild) reduction of viral titres and that the very same SLRs appear degraded (NDP52, TAX1BP1) or translocated to the nucleus (p62). Nuclear translocation presumably interferes with the ability of p62 to antagonize vaccinia replication. The authors report that one of the two known nuclear translocation signals in p62 (NLS2) controls nuclear translocation in vaccinia virus infected cells. They furthermore provide some evidence for two virus encoded kinases (F10 and B1) to be involved in the translocation of p62.

It is well known that autophagy antagonizes pathogen invasion and that many viruses either evade / antagonize autophagy or even take advantage of autophagy components to promote their replication. The current manuscript provides further detail to what we know about Vaccinia -

autophagy interactions. However, not all data are entirely conclusive with the authors' deductions.

1. The authors conclude that VACV 'must have an intrinsic ability to overcome SLR-mediated restriction at native expression levels.' Such conclusion cannot be drawn from the overexpression experiment in Fig1.

Thank you for pointing this out we have rephased to read “providing an indication that VACV might have an intrinsic ability to overcome SLR-mediated restriction”.

2. The authors claim translocation of p62 into the nucleus (Fig2B). Are all images taken at the same settings? How representative are these images - quantification needed, for example percentage of p62+ nuclei.

Yes, all images have been taken at the same setting and displayed as z-stack-maximum projections. The apparent intensity variation is because p62 is not degraded but changes its localization from cytoplasmic dispersed to compact nuclear.

We have quantified both bulk p62 nuclear re-localization over time by western blot (Fig 2C) and p62 nuclear re-localization relative to uninfected cells over time (Fig 2D). We specifically chose this analysis over a binary nuclear/cytoplasmic quantification because in uninfected cells p62 is often evenly dispersed between the nucleus and cytoplasm. By assessing the “amount/signal” of p62 in the nucleus we can confidently say that nuclear p62 levels increase upon VACV infection.

We have also now added images of p62 nuclear re-localization during VACV infection of primary macrophages (Fig S2). These lower magnification images provide an additional overview of the effect with nearly all VACV infected cells displaying p62 nuclear re-localization.

3. If p62 translocates into the nucleus when judged by immunofluorescence (Fig2B), why is the amount of nuclear p62 not changing when judged by Western blot (Fig2C)? If p62 translocates into the nucleus via NLS2, why is there no p62 present in infected cells expressing p62 NLS2 mut at 8h p.i. (Fig3B)

We are not sure what the reviewer is referring to. In the western blot referred to by the reviewer, the amount of nuclear p62 (4th western from top) steadily increases over time up to 18 hpi (please refer to quantification of nuclear vs. cytoplasmic p62 (Fig 2C-right panel). This result is consistent with the kinetic imaging data presented in Fig 2B and the image-based quantification data in 2D.

With regard to Fig 3B, the p62 NLS mutants are expressed at low levels. All images have been taken at the same settings. We did not adjust the imaging conditions to avoid mis-representing the data. In support of the imaging data we have now performed nuclear/cytoplasmic fractionation on cells transiently expressing the p62 WT and NLS mutants left uninfected, or infected with VACV. Consistent with the imaging data we see increased nuclear re-localization of WT and NLS1 at 4 and 8 hpi and no increase in nuclear re-localization of p62 NLS2. Lower expression levels of p62 NLS2 were also observed in these experiments.

Of note, when we enhance the p62 signal by staining for the EGFP tag on these constructs, p62 NLS2 is visibly localized to VACV replication sites at 8 hpi. (See updated Fig 3F).

4. The authors should provide information how often experiments were repeated, and whether repeats represent technical or biological replicates. Statistical analysis missing from some Figures

(for example 1C) and incorrect in many others. I suggest to analyze data by ANOVA, not t-tests, for multiple comparisons (for example 1D, 2D, 3F, 4A, 4B, 4C).

At the end of each figure legend the indicated number of replicates is indicated. All mentions of “replicates” in the figure legends and materials and methods in the manuscript refer to biological replicates.

Figure 1C is quantified in 1D. We have been through the figures suggested by the reviewer. As we are comparing 2 populations in the majority, we have maintained analysis by t-test. We have applied ANOVA to Figs. 2D and 4A, where multiple populations are compared.

5. Fig4C - shades of blue difficult to distinguish. When suggesting a role for F10 and B1, it really would be good if the authors had knockouts (and possibly double knockouts) available, as data would be much more trustworthy. Ideally complemented with wt and kinase dead mutants. Short of this, a single siRNA is insufficient to convince the reader, particularly if not complemented.

We have changed the shades of blue in Fig 4C and Fig 4D to make them more readily distinguishable.

We agree with the reviewer that KOs would be the ideal experiment. Unfortunately, generating KOs of B1, F10, or B1/F10 is technically impossible. Both B1 and F10 are essential VACV kinases. Deletion of B1 prevents VACV replication while deletion of F10 blocks VACV morphogenesis and virion formation. One cannot generate these viruses.

We have previously demonstrated the power of siRNA for the depletion of VACV genes (Kilcher et al., 2014). As opposed to cell lines, the virus is genetically clonal, and the use of single validated siRNAs is routine. Having already demonstrated that the siF10 used here depletes F10 protein during the course of VACV infection (Fig. S5) the addition of another siF10 adds nothing to this experiment.

6. Fig4C - The authors comment only on the 8h time point, where they report a synergistic effect of F10 and B1. Surprisingly, at 4h pi, the late kinase F10 has a stronger phenotype than at 8h and the synergism of F10 with the early kinase B1 is weaker than at 8h pi. Why would F10 and B1, known to act early and late in the viral life cycle, respectively, appear to antagonize autophagy at opposite time points (i.e. late and early)?

Thank you for pointing this out. We suspect this has to do with depletion levels. While the amount of siRNA transfected remains constant the expression levels of F10 (a late gene) increase dramatically between 4hpi and 8hpi (Punjabi & Traktman, 2005). Thus, at early times post infection there would be less F10 than at late times, so the effect of the siRNA on its depletion is likely to be stronger. Regarding synergism of B1 and F10, the Δ B1 virus is propagated in a complementing cell line that supplies B1 kinase in trans. As B1 is packaged in virions, it reasons that Δ B1 virions bring B1 into cells with them, despite not being able to express new B1 from incoming viral genomes. We suspect the differences seen between B1 dependence at 4 and 8 hpi reflect this.

Reviewer #3 (Comments to the Authors (Required)):

This manuscript provides insights into the mechanisms by which vaccinia virus (VACV) alters host cell xenophagy as a mechanism to promote replication. The authors show that infection of HeLa cells with VACV induces the specific shuttling of p62 to the nucleus very early in the viral life cycle (~2hrs post-infection). The authors go on to show that VACV infection induces the phosphorylation of p62, which the authors propose occurs due to the activity of a VACV virally-encoded kinase. Overall, the study provides new insights into the mechanisms by which VACV evades the host autophagic

pathway and points to the role of VACV kinases in this process. The data are well-presented, the manuscript is well-written, and the conclusions are supported by data overall. Despite enthusiasm for the study, I do think there are instances in which additional experimentation could enhance the rigor of the work and/or provide additional clarification on key points.

1. It would appear that most of the work focuses on the overexpression of p62. It would therefore seem quite important to assess the impact of VACV infection on endogenous localization of p62.

The data in Figs 1C, D, E and Fig 2B-D is investigating endogenous p62. We have now indicated this in the figure legends in instances where it was not already indicated.

2. All of the work was performed in HeLa cells, which have an altered autophagic pathway given their nature. The study would be improved by the inclusion of additional data from other cell types (preferably primary in nature) to support this phenomenon in other cells.

We have now assessed p62 nuclear re-localization in primary human monocyte-derived macrophages infected with VACV. Consistent with our findings in HeLa cells, VACV infection causes the re-localization of p62 from the cytoplasm to the nucleus in primary macrophages. This data is now included in the manuscript in Fig S2.

3. In Figure 2A, the authors conclude that VACV avoids p62 degradation early in infection. They base this on immunofluorescence localization studies from cells fixed at various times post-infection. As this would seem to be a key point in the manuscript, the authors should consider live-cell imaging to better resolve these early events as they may be transient in nature.

We have now performed live-cell imaging of EGFP-p62 transfected cell infected with VACVmCh-A5. Imaging was performed from 1hpi-2.2hpi, the time at which VACV fusion has occurred and viral cores are found in the cytoplasm (Rizopoulos et al., 2015). Consistent with our fixed imaging data, we found instances of VACV mCh-A5 virions within EGFP-p62 positive vesicular structures. This data has been included in the revised manuscript as Fig 2A and Fig S2B. The original fixed imaging data has been shifted to Fig S2A.

4. The effects shown in Figure 3F are quite modest. Thus, the authors may wish to alter their conclusions to better reflect these data.

We have toned down the conclusions of this result to be more in line with the data.

5. The authors use BIRB796 to inhibit to inhibit p38 MAPK. More specific means to genetically silence or ablate expression of p38 would better support the lack of a direct link between this molecule and p62 phosphorylation and retargeting.

We have now added a p38 δ -depletion experiment to the manuscript (Fig. S5). We saw no impact on p62 phosphorylation when p38 δ -depleted cells were starved, or infected with VACV (Fig S5). These results suggest that in our system p38 δ does not play a major role in starvation induced p62 phosphorylation. Together with the BIRB796 data, these results strongly suggest that VACV does not act through p38 to phosphorylate p62.

References:

- Kilcher, S., Schmidt, F. I., Schneider, C., Kopf, M., Helenius, A., & Mercer, J. (2014). SiRNA screen of early poxvirus genes identifies the AAA+ ATPase D5 as the virus genome-uncoating factor. *Cell Host and Microbe*, *15*(1), 103–112. <https://doi.org/10.1016/j.chom.2013.12.008>
- Nousiainen, M., Silljé, H. H. W., Sauer, G., Nigg, E. A., & Körner, R. (2006). Phosphoproteome analysis of the human mitotic spindle. *Proceedings of the National Academy of Sciences of the United States of America*, *103*(14), 5391–5396. <https://doi.org/10.1073/pnas.0507066103>
- Novy, K., Kilcher, S., Omasits, U., Bleck, C. K. E., Beerli, C., Vowinckel, J., Martin, C. K., Syedbasha, M., Maiolica, A., White, I., Mercer, J., & Wollscheid, B. (2018). Proteotype profiling unmasks a viral signalling network essential for poxvirus assembly and transcriptional competence. *Nature Microbiology*, *3*(5), 588–599. <https://doi.org/10.1038/s41564-018-0142-6>
- Olsen, J. V., Blagoev, B., Gnad, F., Macek, B., Kumar, C., Mortensen, P., & Mann, M. (2006). Global, in vivo, and site-specific phosphorylation dynamics in signaling networks. *Cell*, *127*(3), 635–648. <https://doi.org/10.1016/j.cell.2006.09.026>
- Pankiv, S., Lamark, T., Bruun, J. A., Øvervatn, A., Bjørkøy, G., & Johansen, T. (2010). Nucleocytoplasmic shuttling of p62/SQSTM1 and its role in recruitment of nuclear polyubiquitinated proteins to promyelocytic leukemia bodies. *Journal of Biological Chemistry*, *285*(8), 5941–5953. <https://doi.org/10.1074/jbc.M109.039925>
- Punjabi, A., & Traktman, P. (2005). Cell Biological and Functional Characterization of the Vaccinia Virus F10 Kinase: Implications for the Mechanism of Virion Morphogenesis. *Journal of Virology*, *79*(4), 2171–2190. <https://doi.org/10.1128/jvi.79.4.2171-2190.2005>
- Rizopoulos, Z., Balistreri, G., Kilcher, S., Martin, C. K., Syedbasha, M., Helenius, A., & Mercer, J. (2015). Vaccinia Virus Infection Requires Maturation of Macropinosomes. *Traffic*, *16*(8), 814–831. <https://doi.org/10.1111/tra.12290>
- Rochester, S. C., & Traktman, P. (1998). Characterization of the Single-Stranded DNA Binding Protein Encoded by the Vaccinia Virus I3 Gene. In *JOURNAL OF VIROLOGY* (Vol. 72, Issue 4). <https://journals.asm.org/journal/jvi>
- Yanagawa, T., Yuki, K., Yoshida, H., Bannai, S., & Ishii, T. (1997). Phosphorylation of A170 stress protein by casein kinase II-like activity in macrophages. *Biochemical and Biophysical Research Communications*, *241*(1), 157–163. <https://doi.org/10.1006/bbrc.1997.7783>

January 17, 2024

RE: JCB Manuscript #202104129R

Prof. Jason Mercer
University of Birmingham
Institute of Microbiology and Infection
Edgbaston
Birmingham B15 2TT
United Kingdom

Dear Prof. Mercer:

Thank you for submitting your revised manuscript entitled "Vaccinia virus subverts xenophagy through phosphorylation and nuclear targeting of p62". The paper has been seen again by two of the original reviewers (reviewer #3 was unavailable to re-review), both of whom have no other experimental requests. Therefore, we would be happy to publish your paper in JCB pending final revisions necessary to meet our formatting guidelines (see details below).

As you will see, reviewer #1 acknowledges your reasons for not being able to experimentally address some of the original reviewer concerns but feels that, because of this, the consequences of p62 trafficking to the nucleus remains unclear. Thus, please be sure to appropriately tone-down the associated conclusions in the paper and try to make clear the limitations and caveats of the study.

A. MANUSCRIPT ORGANIZATION AND FORMATTING:

1) Text limits: Character count for Reports is < 20,000, not including spaces. Count includes the abstract, introduction, results & discussion, and acknowledgments. Count does not include title page, materials and methods, figure legends, references, tables, or supplemental legends. You are somewhat over this limit but we will be able to give you the extra space this time. However, please do try to be as concise as possible in your revisions.

****Please also note that in JCB Reports, the "Results" and "Discussion" sections are combined into a single "Results and Discussion" section so you will need to rewrite those sections to make this change.****

2) Figure formatting: Scale bars must be present on all microscopy images, including inset magnifications. Molecular weight or nucleic acid size markers must be included on all gel electrophoresis. Please be sure to add molecular weight marker(s) to the p-p62 blots in figure 4A-C and in Supp figures 5 and 6.

3) Statistical analysis: Error bars on graphic representations of numerical data must be clearly described in the figure legend. The number of independent data points (n) represented in a graph must be indicated in the legend. Statistical methods should be explained in full in the materials and methods. For figures presenting pooled data the statistical measure should be defined in the figure legends. Please also be sure to indicate the statistical tests used in each of your experiments (both in the figure legend itself and in a separate methods section) as well as the parameters of the test (for example, if you ran a t-test, please indicate if it was one- or two-sided, etc.).

****Also, since you used parametric tests in your study (e.g. t-tests, ANOVA, etc.), you should have first determined whether the data was normally distributed before selecting that test. In the stats section of the methods, please indicate how you tested for normality. If you did not test for normality, you must state something to the effect that "Data distribution was assumed to be normal but this was not formally tested."****

4) Materials and methods: Should be comprehensive and not simply reference a previous publication for details on how an experiment was performed. Please provide full descriptions (at least in brief) in the text for readers who may not have access to referenced manuscripts. The text should not refer to methods "...as previously described."

5) Please be sure to provide the sequences for all of your primers/oligos and RNAi constructs in the materials and methods. You must also indicate in the methods the source, species, and catalog numbers (where appropriate) for all of your antibodies.

6) Microscope image acquisition: The following information must be provided about the acquisition and processing of images:

- a. Make and model of microscope
- b. Type, magnification, and numerical aperture of the objective lenses
- c. Temperature
- d. imaging medium
- e. Fluorochromes
- f. Camera make and model
- g. Acquisition software
- h. Any software used for image processing subsequent to data acquisition. Please include details and types of operations involved (e.g., type of deconvolution, 3D reconstitutions, surface or volume rendering, gamma adjustments, etc.).

7) References: There is no limit to the number of references cited in a manuscript. References should be cited parenthetically in the text by author and year of publication. Abbreviate the names of journals according to PubMed.

8) Supplemental materials: There are strict limits on the allowable amount of supplemental data. Reports may usually have up to 3 supplemental figures. At the moment, you have 6 supp. figures. While we agree that all of this data is necessary for the paper, we feel that it can be presented more concisely. For example, even though they describe different experiments and different aspects of the study, supp. figures 2 and 3 can be combined into a single three-panel figure. Similarly, it should be possible to combine supp. figures 4, 5, and 6 into a single five-panel figure. If you do this, you will be able to fit all the data into three supplementary figures.

****Please be sure to correct the callouts in the text to reflect these changes.****

Please also note that tables, like figures, should be provided as individual, editable files. A summary of all supplemental material (that is, in addition to the supplementary figure legends) should appear at the end of the Materials and methods section. Please see any recent JCB paper for an example of this.]

9) eTOC summary: A ~40-50 word summary that describes the context and significance of the findings for a general readership should be included on the title page. The statement should be written in the present tense and refer to the work in the third person. It should contain "First author name(s) et al..." to match our preferred style.

10) Conflict of interest statement: JCB requires inclusion of a statement in the acknowledgements regarding competing financial interests. If no competing financial interests exist, please include the following statement: "The authors declare no competing financial interests." If competing interests are declared, please follow your statement of these competing interests with the following statement: "The authors declare no further competing financial interests."

11) A separate author contribution section is required following the Acknowledgments in all research manuscripts. All authors should be mentioned and designated by their first and middle initials and full surnames. We encourage use of the CRediT nomenclature (<https://casrai.org/credit/>).

12) ORCID IDs: ORCID IDs are unique identifiers allowing researchers to create a record of their various scholarly contributions in a single place. Please note that ORCID IDs are now ***required*** for all authors. At resubmission of your final files, please be sure to provide your ORCID ID and those of all co-authors.

13) Journal of Cell Biology now requires a data availability statement for all research article submissions. These statements will be published in the article directly above the Acknowledgments. The statement should address all data underlying the research presented in the manuscript. Please visit the JCB instructions for authors for guidelines and examples of statements at (<https://rupress.org/jcb/pages/editorial-policies#data-availability-statement>).

B. FINAL FILES:

****It is JCB policy that if requested, original data images must be made available to the editors. Failure to provide original images upon request will result in unavoidable delays in publication. Please ensure that you have access to all original data images prior to final submission.****

****The license to publish form must be signed before your manuscript can be sent to production. A link to the electronic license to publish form will be sent to the corresponding author only. Please take a moment to check your funder requirements before choosing the appropriate license.****

Thank you for your attention to these final processing requirements. Please revise and format the manuscript and upload materials within 7-14 days.

Thank you for this interesting contribution, we look forward to publishing your paper in Journal of Cell Biology.

Sincerely,

Craig Roy, PhD
Monitoring Editor
Journal of Cell Biology

Tim Spencer, PhD
Executive Editor
Journal of Cell Biology

Reviewer #1 (Comments to the Authors (Required)):

The authors have revised the manuscript to address reviewer concerns. It does seem clear that VACV infection causes p62 to relocate to the nucleus, but the consequences of this remain unclear.

The authors nicely explain why they cannot address some of the key weaknesses in the manuscript.

- 1) The authors cannot KO the F10 kinase proposed to phosphorylate p62 and thus cannot clearly determine if other kinases are mediating p62 phosphorylation in Fig. 4C.
- 2) The authors explain why they cannot over express F10, B1 and Hi kinases preventing them from testing important conclusions by this technique.
- 3) For Rev. 1 point 5, the authors explain the "complex nature of p62" obviates this important experiment.
- 4) For Rev. 1 point 6 the authors explain why this "key experiment" is technically impossible.

Overall, the authors moderate their conclusions and state on page 7, "VACA may shunt p62 to the nucleus in an attempt to prevent autophagic degradation of nascent virions".

Reviewer #2 (Comments to the Authors (Required)):

The authors have addressed all questions. I have no further comments.